# Self-Supervised Selective-Guided Diffusion Model for Old-Photo Face Restoration

**Wenjie Li[1], Xiangyi Wang[1], Heng Guo[1]\*, Guangwei Gao[2], Zhanyu Ma[1]**
[1]Beijing University of Posts and Telecommunications
[2]Nanjing University of Science and Technology
{cswjli, guoheng, mazhanyu}@bupt.edu.cn   csggao@gmail.com

## Abstract

Old-photo face restoration poses significant challenges due to compounded degradations such as breakage, fading, and severe blur. Existing pre-trained diffusion-guided methods either rely on explicit degradation priors or global statistical guidance, which struggle with localized artifacts or face color. We propose Self-Supervised Selective-Guided Diffusion (SSDiff), which leverages pseudo-reference faces generated by a pre-trained diffusion model under weak guidance. These pseudo-labels exhibit structurally aligned contours and natural colors, enabling region-specific restoration via staged supervision: structural guidance applied throughout the denoising process and color refinement in later steps, aligned with the coarse-to-fine nature of diffusion. By incorporating face parsing maps and scratch masks, our method selectively restores breakage regions while avoiding identity mismatch. We further construct VintageFace, a 300-image benchmark of real old face photos with varying degradation levels. SSDiff outperforms existing GAN-based and diffusion-based methods in perceptual quality, fidelity, and regional controllability. Code link: `https://github.com/PRIS-CV/SSDiff`.

## 1 Introduction

Recent image restoration methods [1, 2, 3] excel in generic scenes but struggle with facial regions, may causing misaligned features. To address this, face restoration [4, 5] has been extensively studied, particularly in blind face restoration (BFR) [6, 7, 8, 9, 10, 11] for real scenarios. However, the restoration of old face photographs remains relatively underexplored, as such low-quality (LQ) images often suffer from compounded degradations, including fading, breakage, and severe blurring, which pose challenges for constructing large-scale paired datasets for supervised learning.

Although existing learning-based BFR methods, such as GPEN [7], CodeFormer [9], DifFace [10], and DiffBIR [11], can alleviate blurring in old face photos, they struggle to address degradations like breakage and color fading that are not present in the training data, as illustrated in Fig. 1. Recently, the powerful generative capability of diffusion priors [12, 13] has opened up new possibilities for face restoration. By designing effective guidance strategies for pre-trained diffusion models, the denoising trajectory can be steered toward task-specific objectives, enabling adaptation to various zero-shot restoration tasks [12, 14, 15, 16] in a train-free manner. Such flexibility and low cost make pre-trained diffusion-guided methods well-suited for old-photo face restoration.

Nevertheless, existing pre-trained diffusion-guided methods [14, 16] for old-photo face restoration are limited by rigid, pre-defined priors. For example, DDNM [14] constructs a closed-form solution space using the pseudo-inverse of a linear degradation model and adjusts the denoising trajectory through backpropagation. However, as shown in Fig. 1, degradation in old photos is complex and

---

*Corresponding Author.

39th Conference on Neural Information Processing Systems (NeurIPS 2025).

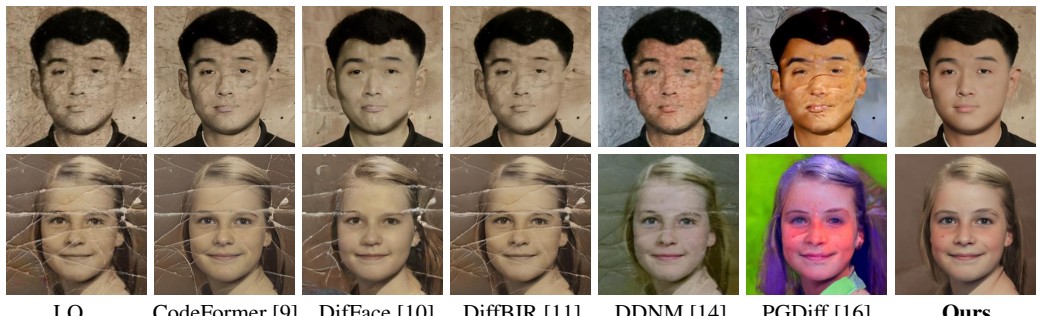

| LQ | CodeFormer [9] | DifFace [10] | DiffBIR [11] | DDNM [14] | PGDiff [16] | **Ours** |

Figure 1: Existing face restoration methods face challenges with complex degradations in old face photos, which may result in residual breakage or unnatural facial colors. In contrast, our method restores sharp facial structures and natural facial color, and no visible breakage regions.

nonlinear, resulting in DDNM only being able to process old photos with relatively simple degradation. PGDiff [16] improves generalization by introducing pre-defined global attribute priors, such as color statistics and smoothing semantics. However, it faces two key limitations: *i)* smoothing semantic priors is less robust to large breakage regions and may cause the model to collapse facial structure consistency during early sampling, leading to noticeable artifacts. *ii)* enforcing global color statistics (*i.e.*, matching the average mean and variance of color channels in FFHQ [17] dataset) may induce sampling bias toward local extrema, resulting in significant image color inhomogeneity.

Motivated by these limitations, we investigate the generative behaviors of pre-trained diffusion models under different guidance and discover a key insight. **Our Insight:** As shown in Fig. 2, we observe that, under weak guidance from a degraded input with appropriate $s$, a pre-trained generative face diffusion model can produce face images that, despite deviating from the original identity (low feature IOU, *e.g.*, eyes, mouth, *etc*), exhibit structurally similar facial contours (high contour IOU), natural color tones (saturation distribution close to HQ faces in FFHQ), and effective restoration in breakage regions (low edge strength variation). This motivates us to use such images as pseudo-references to guide the restoration of facial colors and breakage regions. Meanwhile, facial components like eyes and mouth are not involved in guiding features to prevent low-fidelity facial results.

Building on this insight, we introduce Self-Supervised Selective-Guided Diffusion (SSDiff), a training-free framework that performs self-supervision by using pseudo-references generated under weak guidance. To fully exploit the guidance potential of pseudo-references, SSDiff introduces a staged guidance scheme aligned with the coarse-to-fine nature of diffusion: restoration-oriented guidance, including covering both structure-aware and breakage completion, is applied throughout the process, while color refinement is introduced in later steps, as discussed in Section 3.2. Additionally, we integrate face parsing maps and scratch masks obtained from inputs to selectively guide breakage completion and face coloring without disturbing identity-sensitive features. Through our design, our restorations exhibit high IOU of face contour and components, smooth breakage regions, and consistent color saturation, as shown in Fig. 2. Therefore, SSDiff produces high-quality results even under complex degradations, as shown in Fig. 1. Moreover, benefiting from face parsing-based region selection, SSDiff can also perform region-specific stylized restoration, such as hair and lips.

**Contributions.** **i)** We propose SSDiff, a training-free framework that introduces a reference learning paradigm, where pseudo-references generated under weak guidance from a pre-trained face diffusion model are suitable to steer the reverse process for high-fidelity old-photo face restoration. **ii)** We design a staged, region-specific guidance scheme that combines structure-aware and breakage-aware restoration and late-stage color guidance, enabling robust restoration across diverse degradation types. **iii)** We construct VintageFace, a real-world benchmark of 300 old face photos with varied degradations, and demonstrate that SSDiff consistently outperforms existing GAN-based and zero-shot diffusion methods in perceptual quality and fidelity of restorations.

## 2 Related Work

**Blind Face Restoration.** Blind face restoration aims to recover HQ face images from LQ face images with complex degradations. To enhance visual quality, previous approaches integrate pre-trained

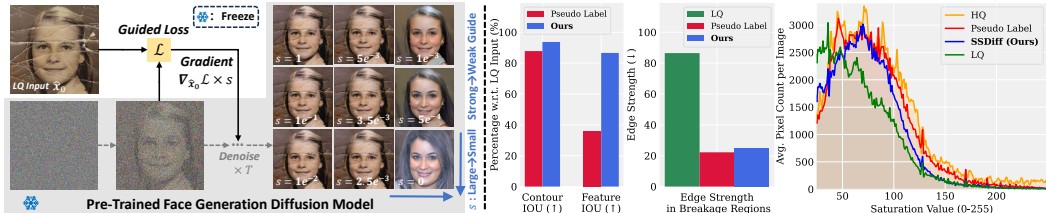

Figure 2: **(Left)** LQ inputs provides supervision via a loss gradient scaled by $s$, aligning reverse diffusion outputs toward it. Strong guidance improves fidelity but harms perceptual quality, while weaker guidance enhances perceptual quality but reduces fidelity. **(Right)** Statistical analysis on our dataset shows that ***pseudo-labels generated under weak guidance ($s = 1e^{-3}$)*** preserve facial contours similar to LQ inputs, exhibit low similarity in semantic regions (e.g., eyes, mouth, nose), and achieve smooth breakage regions with color saturation distributions close to HQ faces.

generative adversarial networks [18] trained on HQ faces into the restoration pipeline. Methods such as GFPGAN [6], GPEN [7], GLEAN [19], SGPN [20], and FREx [21] leverage facial structure cues extracted from degraded inputs and inject them into a pre-trained StyleGAN [17] for restoration. To mitigate the uncertainty of StyleGAN in continuous feature space, approaches like VQFR [8], CodeFormer [9], and DAEFR [22] adopt pre-trained VQGAN [23] to retrieve semantically similar facial features from a discrete codebook. However, GAN-based methods suffer from mode collapse. With the rise of diffusion probabilistic models [24], diffusion-based methods like DR2 [25] employ diffusion models to resist complex degradations, while DifFace [10] models the posterior distribution from LQ to HQ images to generate high-fidelity results. PFStorer [26], PLTrans [27], and Wave-Face [28] further introduce high-frequency features to constrain the denoising process for improved fidelity. Nevertheless, these methods remain limited in handling old-photo face restoration, which may involve breakages and fading that don't appear in training. Our method, situated within the blind face restoration paradigm, is more robust to such challenging degradations in old face photos.

**Diffusion Prior for Restoration.** Methods leveraging diffusion priors for restoration can be broadly categorized into two groups. The first group, including DiffBIR [11], StableSR [29], OSEDiff [30], and OSDFace [31], incorporates pre-trained diffusion models such as Stable Diffusion [13] into the restoration pipeline and fine-tunes controllable modules for generative restoration. These approaches perform well on degradation types seen during training but fail to generalize to unseen ones. The second group focuses on zero-shot restoration by developing efficient guidance strategies for pre-trained diffusion models. Among them, DDRM [32], DDNM [14], GDP [15], and T2I [33] adjust pre-trained diffusion models by estimating a degradation process at each iteration, using fixed linear operators [32, 14, 33] or parameterized degradation models [15] to steer intermediate outputs towards the input LQ images. PGDiff [16] and AGLLDiff [34] guide restoration by modeling desired attributes of HQ images, such as color statistics and structural information from clean datasets. Our method belongs to the second category but differs by using pseudo-references generated through weak guidance instead of explicit attribute modeling. Furthermore, unlike applying global priors, SSDiff selectively guides facial sub-regions in stages, enabling finer control under complex degradation.

## 3 Methodology

### 3.1 Preliminaries

**Denoising Diffusion Probabilistic Models (DDPM).** DDPM generative models [24] consist of two processes: **i)** the forward process progressively adds Gaussian noise $\mathcal{N}$ to the input $\boldsymbol{x}_0$ with a predefined variance schedule $\{\beta_i\}_{t=0}^T$, producing a noisy sample $\boldsymbol{x}_t$ at timestep $t$ according to:

$$q\left(\boldsymbol{x}_t|\boldsymbol{x}_{t-1}\right) = \mathcal{N}\left(\boldsymbol{x}_t; \sqrt{1-\beta_t}\boldsymbol{x}_{t-1}, \beta_t\boldsymbol{I}\right), \tag{1}$$

$$\boldsymbol{x}_t = \sqrt{1-\beta_t}\boldsymbol{x}_{t-1} + \sqrt{\beta_t}\boldsymbol{\epsilon}, \quad \boldsymbol{\epsilon} \sim \mathcal{N}\left(0, \boldsymbol{I}\right). \tag{2}$$

**ii)** The reverse process is parameterized by a learned denoising network $\boldsymbol{\epsilon}_\theta$, typically UNet [35] architecture, which iteratively parameterize the mean value $\boldsymbol{\mu}_\theta\left(\boldsymbol{x}_t, t\right)$ by network $\boldsymbol{\epsilon}_\theta$:

$$p_\theta\left(\boldsymbol{x}_{t-1}|\boldsymbol{x}_t\right) = \mathcal{N}\left(\boldsymbol{x}_{t-1}; \boldsymbol{\mu}_\theta\left(\boldsymbol{x}_t, t\right), \Sigma_\theta(\boldsymbol{x}_t, t)\right), \quad \Sigma_\theta(\boldsymbol{x}_t, t) = \frac{1-\overline{\alpha}_{t-1}}{1-\overline{\alpha}_t}\beta_t\boldsymbol{I}, \tag{3}$$

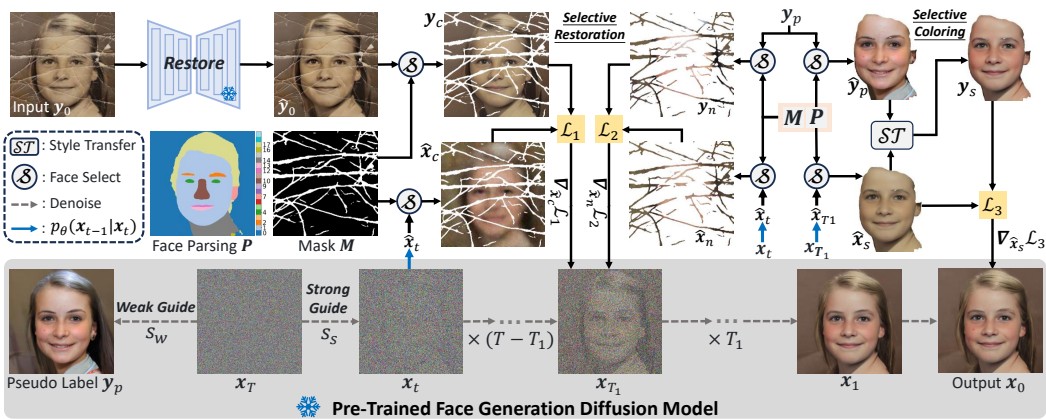

Figure 3: Overview of our method. Face parsing maps $P$ and scratch masks $M$ are estimated from input $y_0$ using a pre-trained face parsing network [36] and a pre-trained scratch detection network [37]. "Weak Guide" refers to the case shown in Fig. 2 at $s = 1e^{-3}$, *i.e.*, $s_w = 1e^{-3}$.

$$\boldsymbol{\mu}_\theta\left(\boldsymbol{x}_t, t\right) = \frac{1}{\sqrt{\alpha_t}}\left(\boldsymbol{x}_t - \boldsymbol{\epsilon}_\theta\left(\boldsymbol{x}_t, t\right)\frac{1 - \alpha_t}{\sqrt{1 - \overline{\alpha}_t}}\right), \quad \alpha_t = 1 - \beta_t, \quad \overline{\alpha}_t = \prod_{i=0}^t \alpha_i. \quad (4)$$

In practice, we can directly estimate $\hat{\boldsymbol{x}}_0 = \boldsymbol{x}_t / \sqrt{\overline{\alpha}_t} - \boldsymbol{\epsilon}_\theta(\boldsymbol{x}_t, t)\sqrt{(1 - \overline{\alpha}_t)/\overline{\alpha}_t}$ from $\boldsymbol{\epsilon}_\theta$.

**Classifier Guidance.** Classifier guidance [12] introduces semantic control into the reverse diffusion process by leveraging gradients from a pre-trained classifier $c_\phi(y|x)$. Instead of training a conditional diffusion model, this approach modifies the denoising step of an unconditional model by shifting the predicted mean based on the classifier gradient $\nabla_x \log c_\phi(y|x)$, evaluated at the intermediate prediction. This guides the generation trajectory toward regions in image space that are more likely to match the target semantics:

$$p_{\theta,\phi}\left(\boldsymbol{x}_{t-1}|\boldsymbol{x}_t, \boldsymbol{y}\right) \approx \mathcal{N}\left(\boldsymbol{\epsilon}_\theta\left(\boldsymbol{x}_t, t\right) + \Sigma_\theta(\boldsymbol{x}_t, t)\nabla_x \log c_\phi(\boldsymbol{y}|\boldsymbol{x})|_{x=\mu_\theta(\boldsymbol{x}_t, t)}, \Sigma_\theta(\boldsymbol{x}_t, t)\right), \quad (5)$$

where the gradient term $g = \nabla_x \log c_\phi(\boldsymbol{y}|\boldsymbol{x})$ acts as a guidance signal that biases the sampling distribution toward the target class $\boldsymbol{y}$, $\Sigma_\theta(\boldsymbol{x}_t, t)$ denotes the degree of diffusion of the sample.

### 3.2 Self-Supervised Selective-Guided Diffusion Model (SSDiff)

As shown in Fig.3, we aim to restore an old face photo LQ input $\boldsymbol{y}_0 \in \mathbb{R}^{C \times H \times W}$, which may suffer from breakage, fading, and blurring, into a HQ face photo output $\boldsymbol{x}_0 \in \mathbb{R}^{C \times H \times W}$. To achieve this, we unify self-supervised loss gradient guidance under a generalized reverse diffusion update, where task-specific losses $\mathcal{L}_1$, $\mathcal{L}_2$ and $\mathcal{L}_3$ as pseudo classifiers to replace traditional classifier gradients, *i.e.*, $\nabla_{\boldsymbol{x}} \log c_\phi(\boldsymbol{y}|\boldsymbol{x}) \propto (\boldsymbol{y} - \boldsymbol{x}) \approx -\nabla_{\boldsymbol{x}}\|\boldsymbol{y} - \boldsymbol{x}\|_2^2$ in Eq.(5). To enable stage-aware supervision, SSDiff integrates two parts: Selective Restoration Guidance, which is applied across the entire reverse trajectory to recover face and breakages, and Selective Coloring Guidance, which is applied in later steps to refine facial color. This temporal separation is empirically motivated by the gradient behavior observed in Fig.4, where image gradient magnitude remains steady in early steps but drops sharply in later steps, indicating

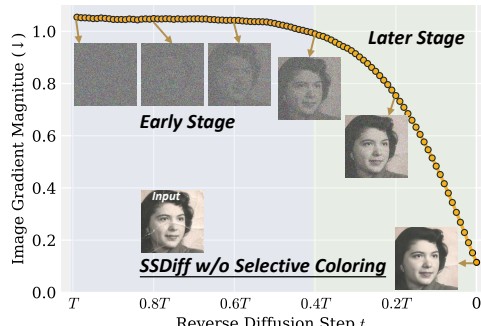

Figure 4: In the early stage, gradient drops are minor, as updates focus on coarse structures like facial contours, which occupy a small region. In the later stage, gradients drop sharply due to refinements in face details, which cover most of a face.

a shift from structural restoration to detail refinement. Our design naturally aligns with the coarse-to-fine denoising curriculum [38, 39] in diffusion models, where early reverse steps reconstruct spatial structures while later steps refine details. Combined with Section 3.1, Algorithm 1 summarizes the full sampling process of SSDiff. Below, we elaborate on its two key components: *Selective Restoration Guidance* and *Selective Coloring Guidance*.

---

**Algorithm 1** : Staged Self-Supervised Sampling in our SSDiff.

---

**Input**: Old face photo $\boldsymbol{y}_0$, face parsing map $\boldsymbol{P}$ and mask $\boldsymbol{M}$ obtained from $\boldsymbol{y}_0$.
**Require**: Pre-trained diffusion model $(\mu_\theta(\boldsymbol{x}_t, t), \Sigma_\theta(\boldsymbol{x}_t, t))$, face selector $\mathcal{S}$, style transfer $\mathcal{ST}$, pseudo label $\boldsymbol{y}_p$ under weak guidance $s_w$, edge magnitude $\mathcal{D}$, strong guidance $s_s$ for restoration.
**Output**: High-quality face photo $\boldsymbol{x}_0$.
$\boldsymbol{x}_T \sim \mathcal{N}(0, \boldsymbol{I}); \boldsymbol{y}_c \leftarrow \mathcal{S}\left(\boldsymbol{y}_p, \boldsymbol{M}\right), \boldsymbol{y}_n \leftarrow \mathcal{S}\left(\boldsymbol{y}_p, \boldsymbol{M}, \boldsymbol{P}\right), \hat{\boldsymbol{y}}_p \leftarrow \mathcal{S}\left(\boldsymbol{y}_p, \boldsymbol{P}\right)$
**for** $t$ from $T$ to 1 **do**
    $\hat{\boldsymbol{x}}_t \leftarrow \frac{1}{\sqrt{\bar{\alpha}_t}}\boldsymbol{x}_t - \sqrt{\frac{1-\bar{\alpha}_t}{\bar{\alpha}_t}}\varepsilon_\theta(\boldsymbol{x}_t, t)$
    $\hat{\boldsymbol{x}}_c \leftarrow \mathcal{S}\left(\hat{\boldsymbol{x}}_t, \boldsymbol{M}\right), \hat{\boldsymbol{x}}_n \leftarrow \mathcal{S}\left(\hat{\boldsymbol{x}}_t, \boldsymbol{M}, \boldsymbol{P}\right)$
    **repeat**
        **if** $t > T_1$ **then**                           $\triangleright$ Stage I: Restoration Guidance
           $\nabla\mathcal{L} = \nabla_{\hat{\boldsymbol{x}}_c}\mathcal{L}_1 + \nabla_{\hat{\boldsymbol{x}}_n}\mathcal{L}_2 = \nabla_{\hat{\boldsymbol{x}}_c}\|\boldsymbol{y}_c - \hat{\boldsymbol{x}}_c\|_2^2 + \nabla_{\hat{\boldsymbol{x}}_n}\|\mathcal{D}(\boldsymbol{y}_n) - \mathcal{D}(\hat{\boldsymbol{x}}_n)\|_2^2$
           $\boldsymbol{x}_{t-1} \sim \mathcal{N}\left(\mu_\theta(\boldsymbol{x}_t, t) - s_s\Sigma_\theta(\boldsymbol{x}_t, t)\nabla\mathcal{L}, \Sigma_\theta(\boldsymbol{x}_t, t)\right)$
        **else**                                $\triangleright$ Stage II: Restoration & Coloring Guidance
           $\hat{\boldsymbol{x}}_{T_1} \leftarrow \frac{1}{\sqrt{\bar{\alpha}_t}}\boldsymbol{x}_{T_1} - \sqrt{\frac{1-\bar{\alpha}_t}{\bar{\alpha}_t}}\varepsilon_\theta(\boldsymbol{x}_t, t); \hat{\boldsymbol{x}}_s \leftarrow \mathcal{S}\left(\hat{\boldsymbol{x}}_{T_1}, \boldsymbol{P}\right), \boldsymbol{y}_s \leftarrow \mathcal{ST}(\hat{\boldsymbol{x}}_s, \hat{\boldsymbol{y}}_p)$
           $\nabla\mathcal{L} = \nabla\mathcal{L} + (\nabla_{\hat{\boldsymbol{x}}_s}\mathcal{L}_3 = \nabla_{\hat{\boldsymbol{x}}_s}\|\boldsymbol{y}_s - \hat{\boldsymbol{x}}_s\|_2^2)$
           $\boldsymbol{x}_{t-1} \sim \mathcal{N}\left(\mu_\theta(\boldsymbol{x}_t, t) - s_s\Sigma_\theta(\boldsymbol{x}_t, t)\nabla\mathcal{L}, \Sigma_\theta(\boldsymbol{x}_t, t)\right)$
        **end if**
    **until** $N$ steps
    $\boldsymbol{x}_{t-1} \sim \mathcal{N}\left(\mu_\theta(\boldsymbol{x}_t, t) - s_s\Sigma_\theta(\boldsymbol{x}_t, t)\nabla\mathcal{L}, \Sigma_\theta(\boldsymbol{x}_t, t)\right)$
**end for**
**return** $\boldsymbol{x}_0$

---

**Selective Restoration Guidance.** Selective restoration guidance is applied throughout the reverse diffusion process, with its main role being to accurately restore the face image and fill in the breakage areas. Given the input $\boldsymbol{y}_0$, we first use an arbitrary pre-trained face restoration model [9, 7] to obtain an initially restored semantics, resulting in an output $\hat{\boldsymbol{y}}_0$ that remains broken. We treat non-broken regions $\boldsymbol{y}_c$ of $\hat{\boldsymbol{y}}_0$ as plausible and use them to guide the frozen pre-trained face generation model to perform reverse diffusion in non-broken regions of faces. We define this part loss $\mathcal{L}_1$ between non-masked face regions $\boldsymbol{y}_c$ of $\hat{\boldsymbol{y}}_0$ and the reverse diffusion-guided output $\hat{\boldsymbol{x}}_c$, formulated as $\mathcal{L}_1 = \|(1 - \boldsymbol{M}) \odot (\boldsymbol{y}_c - \hat{\boldsymbol{x}}_c)\|_2^2$. Its loss gradient $\nabla_{\hat{\boldsymbol{x}}_c}\mathcal{L}_1$ provides supervision to the reverse diffusion process, ensuring faithful restoration in non-broken face regions. $\boldsymbol{M} \in \{0, 1\}$ is a mask obtained by a pre-trained scratch detector [37] from $\boldsymbol{y}_0$, $\odot$ is a Hadamard product.

For breakage regions, previous methods [14, 16] typically set gradients to zero and rely on the pre-trained face generation model to complete them automatically. However, this often results in incomplete completion, especially in areas such as background and skin. To address this issue, we introduce a smoothness-guided loss $\mathcal{L}_2$ which explicitly enforces surface smoothness consistency between pseudo-labels and guided features within breakage regions, mitigating color inconsistency caused by direct guidance. We first use face parsing maps [36] $\boldsymbol{P}$ and scratch masks [37] $\boldsymbol{M}$ derived from input $\boldsymbol{y}_0$ to select plausible breakage guide regions $\hat{\boldsymbol{x}}_n$ and $\boldsymbol{y}_n$ within breakage parts from reverse diffusion and pseudo-labels. This selection focuses on a part of the breakage regions $\boldsymbol{M}_{guide} \subset \boldsymbol{M}$ such as background, skin, and hair, while excluding semantically sensitive areas (*e.g.*, eyes, eyebrows, mouth) that may differ from original identity. Furthermore, we extend ($\mathcal{E}$) the mask of breakages along the horizontal and vertical dimensions of $\boldsymbol{M}_{guide}$ to alleviate incomplete scratch detection [37]. The loss $\mathcal{L}_2$ for the breakage regions is defined as:

$$\mathcal{L}_2 = \|\mathcal{D}\left(\boldsymbol{y}_n, \mathcal{E}\left(\boldsymbol{M}_{guide}\right)\right) - \mathcal{D}\left(\hat{\boldsymbol{x}}_n, \mathcal{E}(\boldsymbol{M}_{guide})\right)\|_2^2, \tag{6}$$

where the function $\mathcal{D}\left(\boldsymbol{y}, \boldsymbol{M}\right) \in \mathbb{R}^{H \times W}$ computes the average per-pixel gradient magnitude within the masked area. Let $\boldsymbol{y}_{i,j}$ denote the pixel value at the spatial location $(i, j)$, then $\mathcal{D}$ is defined as:

$$\mathcal{D}\left(\boldsymbol{y}, \boldsymbol{M}\right) = \left|\boldsymbol{y}_{i,j} - \boldsymbol{y}_{i,j+1}\right|\left(\boldsymbol{M}_{i,j} \odot \boldsymbol{M}_{i,j+1}\right) + \left|\boldsymbol{y}_{i,j} - \boldsymbol{y}_{i+1,j}\right|\left(\boldsymbol{M}_{i,j} \odot \boldsymbol{M}_{i+1,j}\right). \tag{7}$$

The above operations ensure smooth consistency of gradients in breakage regions to restore breakages. Then, we leverage the loss gradient $\nabla_{\hat{\boldsymbol{x}}_n}\mathcal{L}_2$ as a guidance signal during the reverse diffusion process, enabling our model to actively refine breakage regions. For the remaining small portion of breakage, we still allow models to complete them automatically, as pre-trained face generation models generally perform well in restoring semantically critical facial regions such as eyes, eyebrows, and mouth.

Table 1: Quantitative comparison of old-photo face images, which are categorized as simple, medium, and hard based on degradation degree. **Bold** and underlined indicate best and second best results. *Our Appendix provides results on public test sets, including WebPhoto-Test [6], and CelebA-Chlid [6].*

| Type | Metric | GAN-based | | Diffusion-based (Learning) | | Diffusion-based (Train-free) | | |
|---|---|---|---|---|---|---|---|---|
| | | BOPB [37] | Code [9] | DiffFace [10] | DiffBIR [11] | DDNM [14] | PGDiff [16] | Ours |
| **Simple** | FID↓ | 177.19 | 151.01 | 132.14 | 163.47 | 177.82 | 136.73 | **129.13** |
| | BRISQUE↓ | 22.76 | **6.69** | 23.13 | 17.08 | 51.74 | 8.77 | 6.71 |
| | TOPIQ↑ | 0.4444 | 0.6005 | 0.5954 | 0.5474 | 0.3976 | 0.6216 | **0.6494** |
| | MAN-IQA↑ | 0.3124 | 0.3876 | 0.3676 | 0.3568 | 0.2742 | 0.3646 | **0.4005** |
| **Medium** | FID↓ | 176.11 | 189.37 | 142.22 | 198.84 | 173.96 | 142.56 | **128.31** |
| | BRISQUE↓ | 22.54 | 8.29 | 23.45 | 12.27 | 52.78 | 10.22 | **7.29** |
| | TOPIQ↑ | 0.4337 | 0.6123 | 0.5917 | 0.5684 | 0.3929 | 0.6060 | **0.6412** |
| | MAN-IQA↑ | 0.2975 | 0.3789 | 0.3511 | 0.3509 | 0.2721 | 0.3456 | **0.3949** |
| **Hard** | FID↓ | 218.35 | 173.89 | 145.85 | 200.38 | 203.13 | 166.42 | **122.59** |
| | BRISQUE↓ | 28.91 | 10.51 | 26.41 | 14.36 | 54.17 | 13.60 | **8.95** |
| | TOPIQ↑ | 0.3865 | 0.5670 | 0.5450 | 0.5424 | 0.3447 | 0.5747 | **0.5977** |
| | MAN-IQA↑ | 0.2454 | 0.3379 | 0.3159 | 0.3140 | 0.2243 | 0.3159 | **0.3575** |

| LQ | BOPB [37] | CodeFormer [9] | DiffFace [10] | DiffBIR [11] | DDNM [14] | PGDiff [16] | **Ours** |

Figure 5: Qualitative comparisons of old-photo face images with existing methods.

**Selective Coloring Guidance.** Selective coloring guidance is applied only after facial identity features such as eyes, nose, and mouth have stabilized during the reverse diffusion process. At this stage, as shown in Fig.4, identity features are essentially fixed, and remaining steps focus on refining details, which reduces the risk of introducing identity inconsistency through color guidance. Given intermediate images $\boldsymbol{x}_{T_1}$ from the reverse diffusion and pseudo-label $\boldsymbol{y}_p$, we first extract regions related to the skin, including face and neck, using face parsing maps $\boldsymbol{M}$, resulting in $\hat{\boldsymbol{x}}_s$ and $\hat{\boldsymbol{y}}_p$. However, direct supervision from $\hat{\boldsymbol{y}}_p$ to $\hat{\boldsymbol{x}}_s$ is suboptimal due to potential variations in facial components. To address this, we employ a pre-trained color style transfer network [40] to transfer the color style of $\hat{\boldsymbol{y}}_p$ to $\hat{\boldsymbol{x}}_s$ to obtain a color-adapted intermediate result $\boldsymbol{y}_s$, which subsequently guides the remaining reverse diffusion steps for faithful and coherent face coloring. To supervise this process, we define a color consistency loss $\mathcal{L}_3$ between $\hat{\boldsymbol{x}}_s$ and $\boldsymbol{y}_s$, which enforces global color coherence while allowing structural flexibility in regions sensitive to identity. Specifically, $\mathcal{L}_3$ is defined as:

$$\mathcal{L}_3 = \|\boldsymbol{y}_s - \hat{\boldsymbol{x}}_s\|_2^2, \quad \boldsymbol{y}_s = \text{ColorTransfer}\left(\hat{\boldsymbol{x}}_s, \hat{\boldsymbol{y}}_p\right). \tag{8}$$

In this phase, by using the loss gradient $\nabla_{\hat{\boldsymbol{x}}_s}\mathcal{L}_3$ to guide tones and shadows of skin regions, and combining this with the earlier selective restoration guidance, facial coloring realism, and identity preservation are improved while supporting further fine-grained facial detail reconstruction.

**Overall Guidance.** Our guidance strategy is staged to align with the progression of the reverse diffusion. In the early phase of denoising, where $t \in [T_1, T]$, models focus solely on structural restoration. We apply guidance losses $\mathcal{L} = \mathcal{L}_1 + \mathcal{L}_2$, where $\mathcal{L}_1$ encourages fidelity in intact regions and $\mathcal{L}_2$ enforces smoothness in breakage regions. Once the facial identity stabilizes, in the later phase $t \in [0, T_1]$, we introduce a selective coloring loss $\mathcal{L}_3$ and formulate the full loss as $\mathcal{L} = \mathcal{L}_1 + \mathcal{L}_2 + \mathcal{L}_3$.

In this framework, pseudo-labels generated under a weak guidance scale $s_\text{w}$ are used to supervise the actual restoration process, which is conducted under a stronger guidance scale $s_\text{s}$. To ensure coherent interaction between the two stages, we adopt a joint optimization perspective:

$$\mathcal{L} = \arg\min \left( \mathbb{E}\|\boldsymbol{y}_c(s_\text{w}^*) - \hat{\boldsymbol{x}}_c(s_\text{s}^*)\|_2^2 + \mathbb{E}\|\boldsymbol{y}_n(s_\text{w}^*) - \hat{\boldsymbol{x}}_n(s_\text{s}^*)\|_2^2 + \mathbb{E}\|\boldsymbol{y}_s(s_\text{w}^*) - \hat{\boldsymbol{x}}_s(s_\text{s}^*)\|_2^2 \right). \tag{9}$$

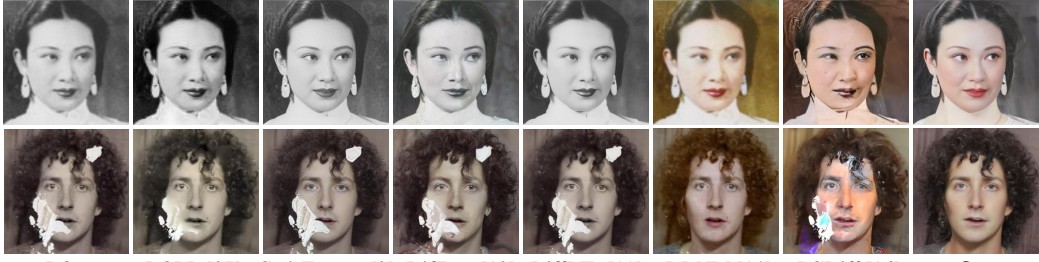

| LQ | BOPB [37] | CodeFormer [9] | DiffFace [10] | DiffBIR [11] | DDNM [14] | PGDiff [16] | **Ours** |

Figure 6: Quantitative comparisons of results on no breakage and large-region breakage images.

Theoretically, an optimal pair of weak and strong guidance scales $(s_\text{w}^*, s_\text{s}^*)$ achieves an equilibrium, where neither under- nor over-guidance dominates, enabling effective perceptual restoration.

# 4 Experiments

## 4.1 Experiments Setting

**Datasets and Metrics.** Our method is **training-free** and does not require a training dataset. For evaluation, we randomly collect 300 old face photographs from the Internet as our benchmark, called VintageFace. All images are cropped and aligned to $512 \times 512$ using the open-source FaceLib library[1]. We further categorize them into three levels of degradation: simple, medium, and hard, based on image quality, 100 images each. Details of the benchmark are provided in our *Appendix*. Since ground-truth is unavailable, we follow previous works [10, 11] and adapt no-reference metrics, like FID [41], BRISQUE [42], TOPIQ [43], and MAN-IQA [44] to evaluate perceptual quality. Furthermore, we provide qualitative results and CLIP-based [45] identity distance to assess restoration fidelity.

**Implementation Details.** We use the pre-trained real-time model BiseNet [36] and the scratch detection model from [37] to obtain face parsing maps and scratch masks from inputs, respectively. Our pre-trained diffusion model is an unconditional denoising network trained on FFHQ [17] datasets, which learns to reconstruct high-quality faces from pure noise over $T = 1000$ steps. We restore breakage facial regions during the first 600 steps, and apply color migration in the remaining 400 steps ($T_1 = 400$). The strong gradient factor of our SSDiff is $s_s = 3.5e^{-3}$. PGDiff [16] is also set to this value for fairness. For the style transfer process shown in Fig. 3, we adopt a pre-trained lightweight model CAP-VSTNet [40], without using its built-in segmentation module for boundary invariance. All experiments are implemented in PyTorch framework on an NVIDIA RTX 4090 GPU.

## 4.2 Comparisons with Existing Methods

We compare our SSDiff with BOPB [37], which is specialized for old photo restoration; Code-Former [9], DiffFace [10], and DiffBIR [11], which are designed for blind face restoration; and DDNM [14] and PGDiff [16], which guide pre-trained diffusion for zero-shot old-photo restoration.

As shown in Table 1, a quantitative comparison of our SSDiff with the above methods demonstrates that our approach not only significantly outperforms existing methods in FID, BRISQUE, and TOPIQ, which reflect image features quality, but also achieves more natural image quality in the cross-modal human perceptual metric MAN-IQA. However, these metrics alone do not fully capture the fidelity of the results. Therefore, Figure. 5 presents qualitative comparisons across three types of old face photographs: simple, medium, and

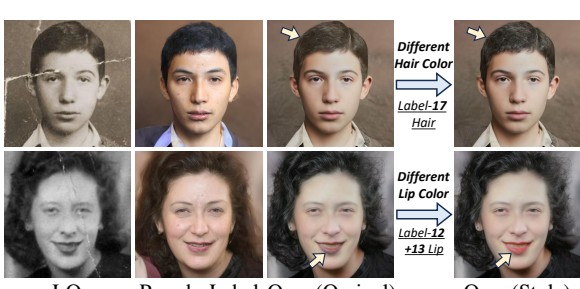

| LQ | Pseudo-Label | Ours (Orginal) | Ours (Style) |

Figure 7: Our method allows stylized restoration of facial components specified by reference to pseudo-labels on inputs.

hard. Our method produces face images with natural color, clear texture, and minimal damage, while maintaining high fidelity and preserving the identity of the input face images. Furthermore, we provide more old-face photo restoration results in our *Appendix*.

---

[1]Facelib: https://github.com/sajjjadayobi/FaceLib

**Evaluation on Diverse Old Face Photos.**    As shown in Fig. 6, we validate the generality of our SSDiff on a broader set of old face photos. This includes no-breakage images where the mask is all zeros, and large-region breakage images where the mask indicates missing regions to be filled. In all scenarios, our method performs robustly, achieving high-quality face coloring and region completion while restoring sharp facial contours. It significantly outperforms existing methods that are limited to a single function, such as coloring, completion, or suffer from overall poor results.

**Region-Specific Stylized Restoration.**    Since the face parsing map is insensitive to scratches and highly structured features of faces, we can reliably select specific facial components for targeted restoration using fixed semantic labels. As shown in Fig. 7, our restored faces exhibit unnatural violet around the lips. This issue can be mitigated by explicitly selecting the lip region via the parsing map and increasing the guidance strength for this component. Furthermore, the same strategy can be applied to adjust hair color, for example, modifying yellowish hair to a dark tone by reference to pseudo-labels. This property allows us to fine-tune facial local components with suboptimal details and achieve more visually appealing face restoration results.

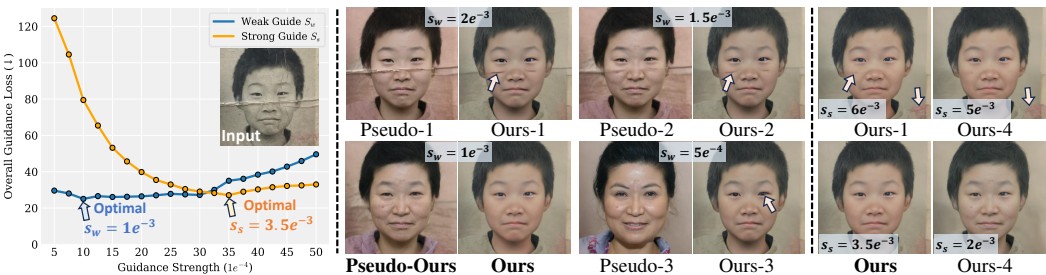

Figure 8: Ablation studies on the effect of different weak guidance factors $s_w$ for generating pseudo labels and strong factors $s_s$ in restoration on results. When $s_s = 1e^{-3}$ and $s_w = 3.5e^{-3}$, our results show the best guidance loss, facial tones, facial details, and fidelity of identity.

## 4.3   Ablation Studies

**Self-Supervised Strategy.**    When the self-supervised strategy is removed, namely when the pseudo-label $y_p$ is no longer used, we disable both the selective restoration guidance and selective coloring guidance associated with $y_p$ in breakage regions. As shown in Table 2, results show a significant drop in visual quality without the self-supervision strategy. Furthermore,

Table 2: Ablation studies of our self-supervised strategy on "medium" type data.

| Self-Supervise | FID↓ | TOPIQ↑ | BRISQUE↓ |
|---|---|---|---|
| ✗ | 145.7 | 0.618 | 7.92 |
| ✔ | **128.3** | **0.641** | **7.29** |

based on Fig. 2, we explore the appropriate range for weak and strong guide factors $s_w$ and $s_s$, which determines the reliability of generated pseudo-labels and results. As shown in Fig. 8, an excessively large $s_w$ leads to incomplete removal of pseudo-label scratches, leaving visible artifacts in restorations. Conversely, an overly small $s$ causes some identity feature mismatch, resulting in distorted eyes and dark facial tones. Thus, we set $s_w = 1e^{-3}$ and $s_s = 3.5e^{-3}$ to obtain best results.

**Selective Guidance.**    We first evaluate the effectiveness of the three guided losses: $\mathcal{L}_1$, $\mathcal{L}_2$, and $\mathcal{L}_3$. As shown in Table 3, all three gradient-based losses contribute to improved restoration. As shown in Fig. 10, visualization results further reveal that $\mathcal{L}_1$ and $\mathcal{L}_2$ enhance the sharpness of facial structures, reducing visible breakage artifacts in old photos. In contrast, $\mathcal{L}_3$ plays a critical role in facial coloring, ensuring a natural tone in results. Furthermore, since $\mathcal{L}_3$ is applied after

Table 3: Ablation studies of our selective guidance strategy on "medium" type data.

| Guidance | FID↓ | TOPIQ↑ | BRISQUE↓ |
|---|---|---|---|
| **Ours** | **128.3** | **0.641** | **7.29** |
| w/o $\mathcal{L}_1$ | 142.8 | 0.602 | 8.18 |
| w/o $\mathcal{L}_2$ | 132.1 | 0.630 | 7.43 |
| w/o $\mathcal{L}_3$ | 136.3 | 0.624 | 7.65 |

the facial structure has stabilized, we investigate the appropriate value for $T_1$. As shown in Fig. 9, letting $T$ denote the total number of diffusion steps, setting $T_1 = 0.4T$ (fluctuating with degradations) stabilizes the facial structure and enables the guided coloring to produce more natural skin tones.

Next, we justify the design of selective restoration. The pseudo-label $y_p$ is not used to guide high-semantic facial components such as the eyes, nose, or mouth, as these regions are identity-sensitive and pseudo-label inconsistencies may degrade fidelity, as shown in Fig. 10. For selective coloring,

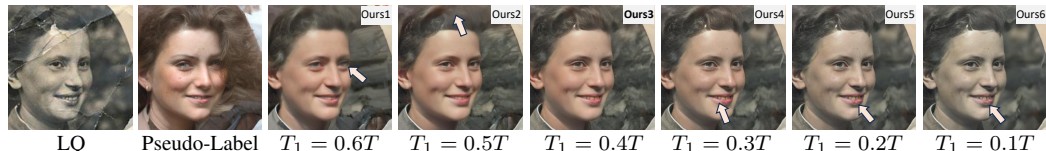

Figure 9: Ablation study on different $T_1$ values in reverse diffusion, *i.e.*, when to start guiding face coloring, on restoration results. Best facial tones and details are observed at $T_1 = 0.4T$.

we examine why guidance is applied only to skin regions, rather than full faces, or why we don't just color transfer. As shown in Fig. 10, due to the limited performance of the pre-trained color transfer network and large color variations of overall faces, the overall direct migration or full face guidance leads to unsatisfactory results. Therefore, we restrict color guidance to skin regions.

**Fidelity of Restoration.** Due to the presence of severely degraded facial features in a part of inputs, it is challenging to reliably evaluate identity consistency across all samples. To ensure meaningful measurement, we select 50 LQ inputs with relatively well-preserved facial features and compute CLIP-based [45] identity feature distance between the restorations and inputs. As shown in Table 4, our method achieves low identity distances, indicating good fidelity in restoring identity-relevant features.

Table 4: Quantitative comparison on identity distances ($\downarrow$) with LQ inputs.

| BOPB [9] | DiffFace [10] | DiffBIR [11] |
|---|---|---|
| 0.2348 | 0.2451 | **0.2223** |
| DDNM [14] | PGDiff [16] | **Ours** |
| 0.2386 | 0.2859 | 0.2280 |

### 4.4 Limitations and Future Works

Our method improves the robustness of face restoration for old photos and performs well even under complex breakage. However, as shown in Fig. 11, when large stained regions are present, our method may mistakenly restore these stains as part of the face, such as skin, leading to unclean results. This issue stems from the diffusion model's tendency to over-rely on surrounding context in severely degraded regions, leading to incorrect restorations when visual cues are ambiguous. In the future, we plan to incorporate fine-grained region-level annotations for such cases to reduce failure modes.

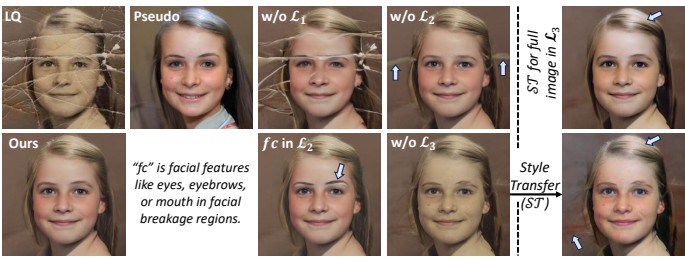

Figure 10: (**Left**) Ablations of $\mathcal{L}_1$, $\mathcal{L}_2$, $\mathcal{L}_3$, and why we can't guide "fc" in breakages in $\mathcal{L}_2$. (**Right**) Ablations of color-guided areas, and why don't we just color style transfer ($\mathcal{ST}$) instead of guiding coloring?

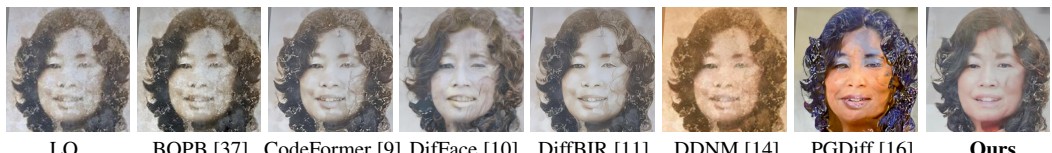

Figure 11: Our Failures. When large stains appear, they may be misinterpreted as a part of facial or hair structures and color by our method, leading to unnatural artifacts and distorted local details.

## 5 Conclusion

In this work, we present SSDiff, a specialized framework for old-photo face restoration that leverages the generative capacity of pre-trained diffusion models through self-supervised pseudo-reference guidance. Unlike existing approaches that rely on explicit degradation models or pre-defined high-quality face attributes, SSDiff guides the denoising process using features selected from generated pseudo-references and facial geometry priors such as parsing maps and scratch masks at different sampling stages. This design enables the selective restoration of plausible facial regions of old photos while maintaining high visual fidelity. Extensive experiments demonstrate that SSDiff not only handles complex old-photo degradations effectively but also enables region-specific stylization of faces, offering a flexible and robust solution for challenging old-photo face restoration tasks.

## 6   Acknowledgments

This work was supported by National Natural Science Foundation of China (Grant No. 62472044, U24B20155, 62225601, U23B2052), Beijing-Tianjin-Hebei Basic Research Funding Program No. F2024502017, Hebei Natural Science Foundation Project No. 242Q0101Z, Beijing Natural Science Foundation Project No. L242025.

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

# Appendix

## A  Details of Our Benchmark

First, as illustrated in Fig. S12, we present our criteria for categorizing our proposed VintageFace benchmark into simple, medium, and severe degradation levels. Specifically, we employ a frozen CLIP model to compute the similarity between each old photo and a textual description of degradation severity. Images are then ranked by their similarity scores and assigned to categories accordingly. To ensure accuracy, we further manually corrected a small number of misclassified samples.

Second, as shown in Fig.S13, we display representative examples from each degradation level in the VintageFace benchmark. These examples demonstrate varying degrees of blurring, fading, and structural damage, and are largely consistent with the classification criteria established in Fig.S12.

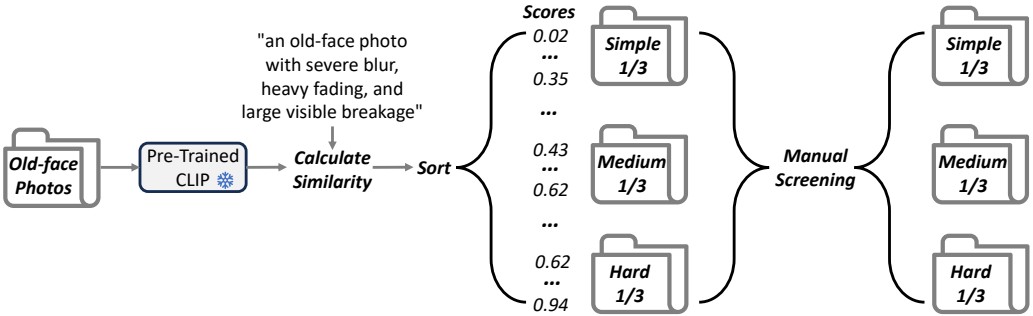

Figure S12: Method for categorizing degradation types into simple, medium, and hard levels in our benchmark VintageFace for testing.

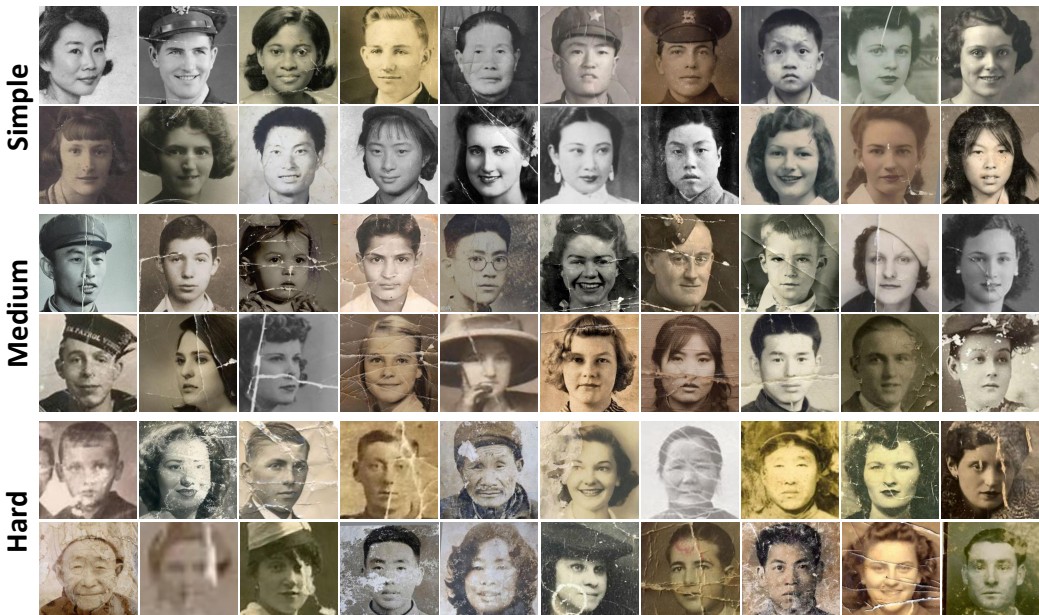

Figure S13: A showcase of representative facial images with varying degradation types from our VintageFace benchmark. The benchmark includes faces across diverse genders, ages, and ethnicities.

## B  More Comparisons

First, we provide additional visual results in the appendix (*e.g.*, Fig.S16, Fig.S17, and Fig.S18) to complement the main text. These figures showcase the restoration performance on old face

Table S5: Quantitative comparison on real-world BFR benchmarks that contain old face photos, like WebPhoto-Test and CelebA-Child. **Bold** and underlined indicate best and second best results.

| Dataset | Metric | GAN-based | | Diffusion-based (Learning) | | Diffusion-based (Train-free) | | |
|---|---|---|---|---|---|---|---|---|
| | | GPEN [7] | Code [9] | DifFace [10] | DiffBIR [11] | DDNM [14] | PGDiff [16] | **Ours** |
| **WebPhoto-Test** | FID↓ | 101.3 | **83.2** | 89.1 | 91.8 | 165.6 | 96.1 | 86.9 |
| | NIQE↓ | 6.326 | 4.705 | 4.831 | 6.069 | 9.259 | 5.117 | **4.406** |
| **CelebA-Child** | FID↓ | 113.0 | 116.2 | 113.1 | 118.9 | 151.2 | 121.0 | **112.7** |
| | NIQE↓ | 4.945 | 4.983 | 4.818 | 5.549 | 6.576 | 5.070 | **4.524** |

photographs across diverse genders, ethnicities (Asian, European-American, and Indian), and age groups. Our method effectively balances perceptual quality and identity preservation: the restored images exhibit minimal artifacts or breakage while maintaining faithful facial identity.

Second, we observe that widely used blind face restoration benchmarks, such as LFW-Test and CelebChild, also include a substantial number of old face photos. However, these differ from our dataset in that they primarily exhibit blurring, with no significant structural damage and limited fading. To demonstrate the generalization and effectiveness of our method, we compare it with state-of-the-art approaches on these two benchmarks. Following previous works [9, 16], we adopt FID and NIQE as evaluation metrics. As shown in Table S5, our method achieves good quantitative results on both benchmarks. Furthermore, we provide visual comparisons in Fig. S19, which reveal that our method not only effectively addresses blurring but also excels at restoring facial color. This perceptual advantage, particularly in color restoration, is not fully captured by quantitative metrics.

Thirdly, VintageFace primarily consists of frontal photos, as portrait photos decades ago were typically studio-based, focusing on clearly capturing facial features, making profile shots rare. Additionally, eyewear was less common, resulting in fewer photos with glasses. Consequently, our data has fewer such samples. Nevertheless, as shown in Fig. S14, SSDiff performs robustly across these scenarios, including glasses, profile shots, and severe degradations, consistently yielding favorable results.

# C   More Ablation

**Robustness of Pre-trained Networks.**   Our SSDiff is generally robust to inaccuracies in external components (face parsing, scratch detection, style transfer). These networks only provide coarse directional signals during reverse diffusion, similar to classifier-guided diffusion, and are not strict constraints. As long as the guidance is not severely misleading, the strong generative prior of the frozen diffusion model dominates reconstruction. To quantify this robustness, as shown in the Table S6, Table S7, and Table S8, we conduct ablations on the Medium type subset:

For parsing map networks, we introduce inaccuracies by replacing the original parsing maps with pseudo-label parsing maps of different strengths $s$, where the resulting errors are even larger than those observed in parsing maps under severe degradations (79% IOU). The resulting IoU with the original parsing map is: for s=2.5e-4, IoU=76% (24% discrepancy); for s=1e-4, IoU=70% (30% discrepancy); For scratch detection networks, we randomly flip 10%, 20%, and 30% of the masks of breakage regions to simulate a situation where some of the breakages have not been detected; For style transfer networks, we weaken the style transfer guidance by reducing the style factor $\alpha$ from 0 to 0.1 and 0.2, slightly affecting color and content.

| | Table S6: Parsing Maps. | | | Table S7: Scratch Masks. | | | | Table S8: Style Transfer. | | |
|---|---|---|---|---|---|---|---|---|---|---|
| | Ours | 76% | 70% | Ours | 10%flip | 20%flip | 30%flip | Ours ($\alpha$=0) | $\alpha$=0.1 | $\alpha$=0.2 |
| FID(↓) | 128.3 | 131.1 | 133.4 | 128.3 | 129.2 | 130.7 | 132.5 | 128.3 | 129.2 | 128.8 |
| MAN-IQA(↑) | 0.395 | 0.391 | 0.382 | 0.395 | 0.391 | 0.379 | 0.381 | 0.395 | 0.396 | 0.392 |
| Face Sim.(↑) | 1 | 0.985 | 0.955 | 1 | 0.974 | 0.953 | 0.937 | 1 | 0.977 | 0.959 |

Here, Face Similarity (range [0, 1]) denotes the cosine similarity between features (extracted with ArcFace [46]) of the perturbed restoration and the original restoration (Ours). These results show that errors in the pre-trained networks are not severe, and the strong generative prior of the diffusion model can propagate the correct cues to other regions, preventing significant performance drops. This demonstrates that SSDiff is robust to these pre-trained networks.

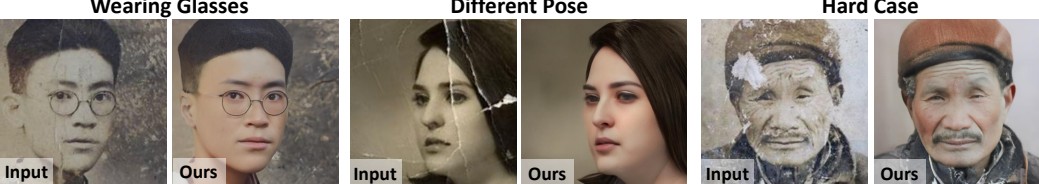

Figure S14: Visualization of SSDiff under wearing glasses, different poses, and severe degradation.

**Latency.** Our method is built upon existing pre-trained diffusion face generation models, where additional inference overhead mainly comes from four components: a simple restore, a face parsing network, a scratch detection network, and a style transfer network. All components are lightweight and are executed at a single denoising step rather than throughout the entire process. Moreover, except for the style migration network, the other three can optionally be pre-processed offline. When all four components are executed online, the average latency for processing a single old face photo is 95 ms on an NVIDIA GeForce RTX 4090. If the three offline-optional components are pre-processed, the average latency is reduced to 24 ms. In contrast, PGDiff [16] requires semantic information extraction at each denoising step, introducing a latency of about 10 s. Therefore, our method only introduces minimal latency to existing pre-trained generation diffusion frameworks.

**Computational Cost.** As shown in Table S9, we further compare the number of Parmas, FLOPs, and inference time of our method with existing diffusion-based face restoration methods [10, 11, 16]. We let the restore be performed offline; our method

Table S9: Quantitative comparison on computational costs with existing diffusion-based BFR methods.

| Costs | DifFace [10] | DiffBIR [11] | PGDiff [16] | **Ours** |
|---|---|---|---|---|
| Parmas↓ | 175.4M | 1717M | 47.7M | **45.4M** |
| FLOPs↓ | 268.8G | 24234G | 127.5G | **120.7G** |

performs excellently. Our method is smaller in terms of the Parmas and FLOPs counts, especially compared to stable diffusion-based methods like DiffBIR [11].

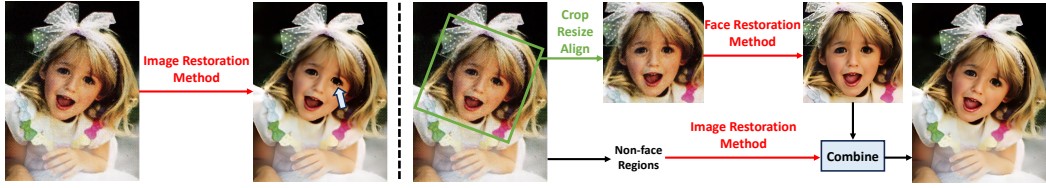

Figure S15: (**Left**) Face images are highly sensitive to artifacts, directly restoring photos containing faces with image restoration methods may results in visually disturbing results. (**Right**) A common strategy involves cropping and aligning facial regions, followed by restoration using face restoration methods, while non-facial regions are enhanced with image restoration methods to ensure visual perception. *Therefore, old photo face restoration holds practical value for old photo restoration.*

# D   Necessity of Old-Photo Face Restoration

While general image restoration methods [1] aim to restore the entire image holistically, we argue that dedicated face restoration [6] is necessary and beneficial, especially in the context of severely degraded old portraits. As shown in Fig. S15, directly applying general real-world image restoration models [47] to facial regions may introduce noticeable artifacts, even when these methods perform reasonably well on background areas. This is because facial regions are typically small in size, contain rich structural priors (*e.g.*, eyes, nose, mouth), and are highly sensitive to local distortions. Artifacts in these regions are particularly perceptible and detrimental to human perception.

Similarly, old face photos suffer from unique degradation patterns such as heavy blurring, fading, and structural damage. Applying global restoration methods [47, 48] to these faces without region-specific modeling frequently leads to distorted identity features or unnatural textures. Therefore, we advocate for face-specific old photo restoration approaches [14, 16] that focus on preserving facial identity and fidelity, while allowing general old photo restoration techniques [37] to handle the surrounding non-facial regions. This targeted strategy ensures high-quality restoration where perceptual sensitivity is highest and complements broader restoration pipelines. Therefore, we respectfully believe the task of old photo face restoration holds specifically practical value.

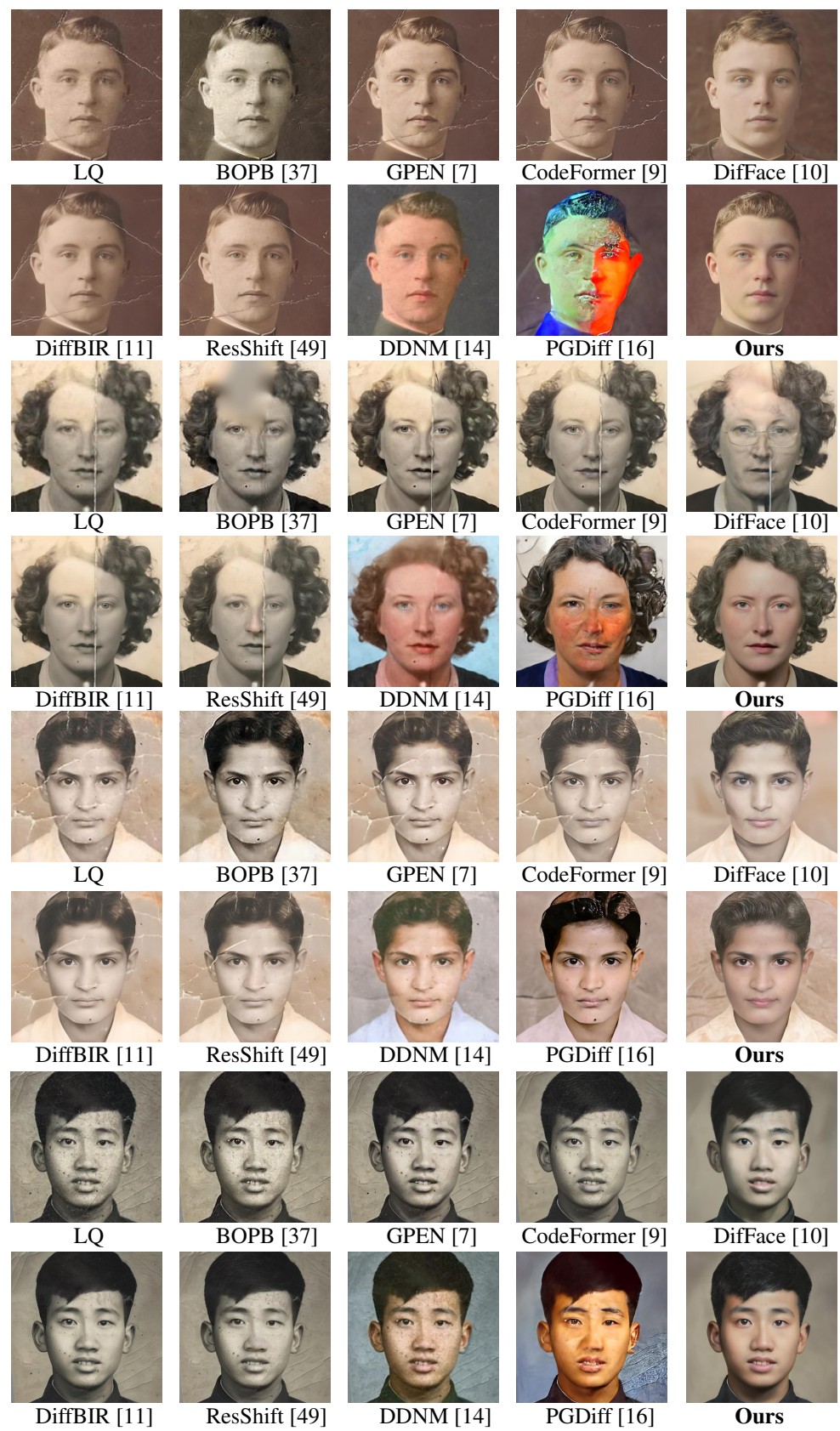

Figure S16: Qualitative comparisons with existing methods on our VintageFace.

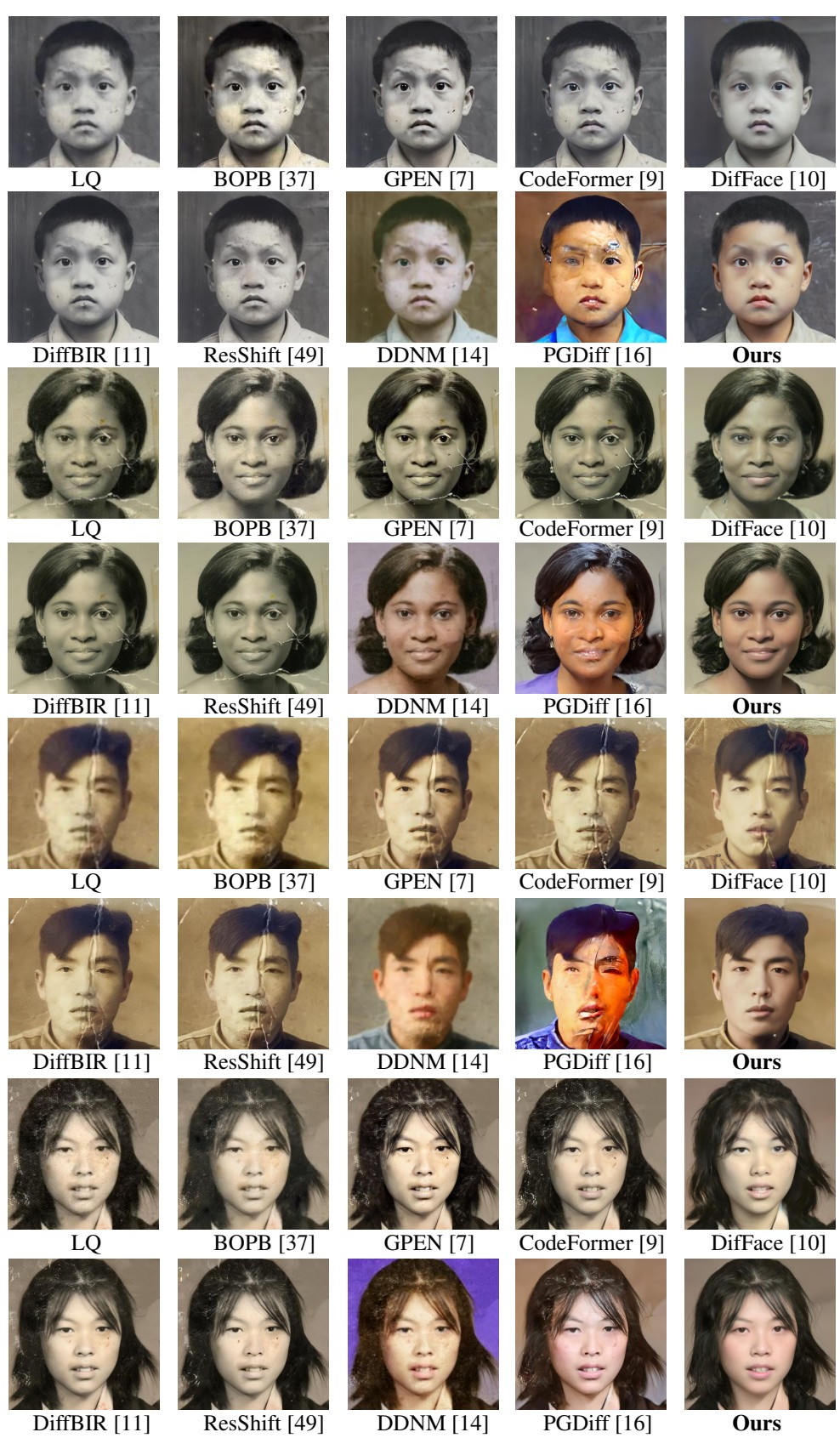

Figure S17: Qualitative comparisons with existing methods on our VintageFace.

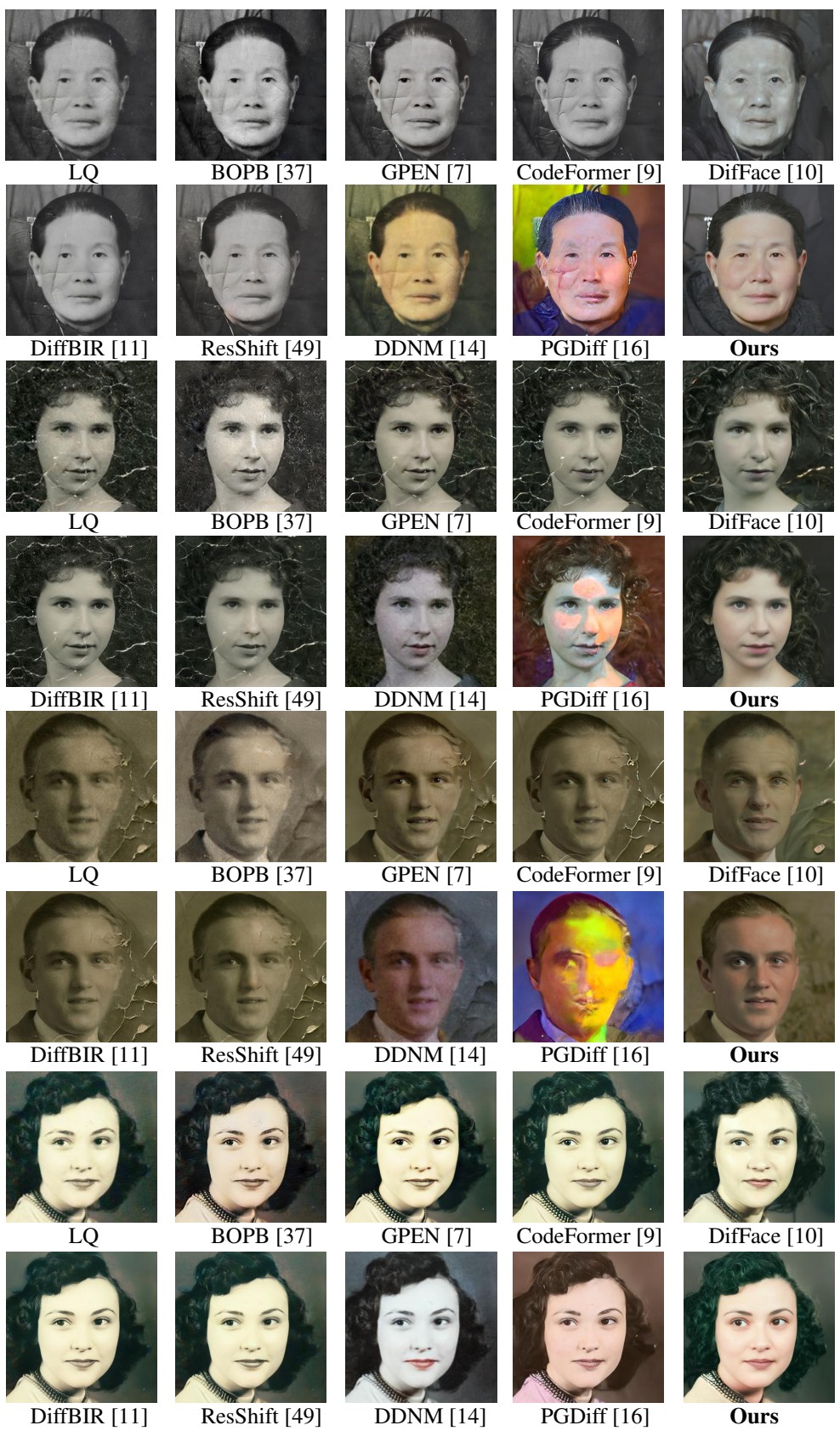

Figure S18: Qualitative comparisons with existing methods on our VintageFace.

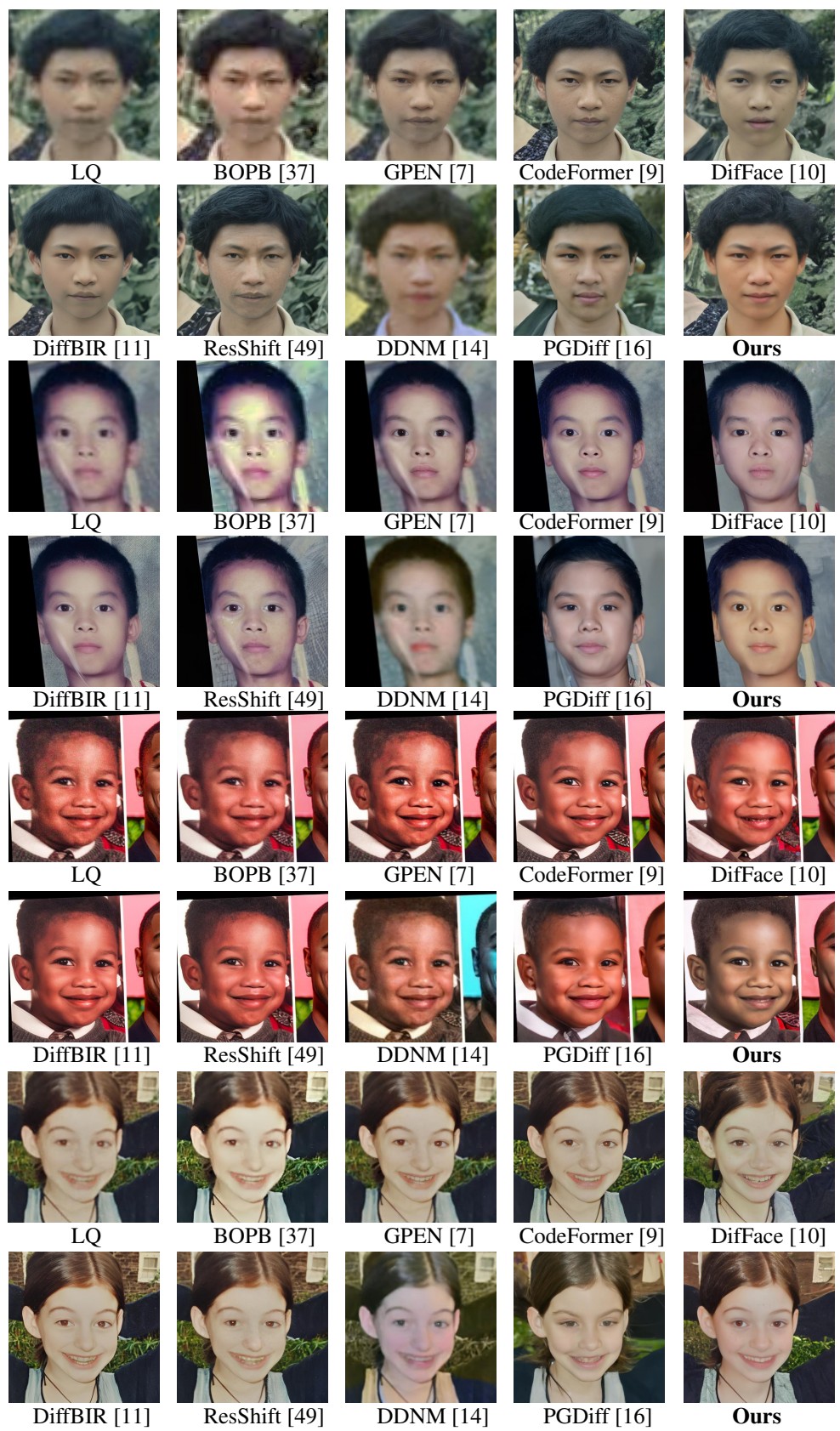

Figure S19: Qualitative comparisons with existing methods on WebPhoto-Test and CelebA-Child.

