# OpenReview forum: "Self-Supervised Selective-Guided Diffusion Model for Old-Photo Face Restoration"
_NeurIPS.cc/2025/Conference — NeurIPS 2025 poster_

### Official Review · Reviewer_pXcG · 2025-06-12

**Clarity:** 3
**Significance:** 3
**Originality:** 3
**Rating:** 6
**Confidence:** 5

**Summary:**

SSDiff combines weak guidance to generate pseudo-labels, and employs selective restoration with selective coloring using structural awareness to extend the paradigm of pre-trained zero-shot diffusion guidance, expanding the new paradigm of zero-shot diffusion guidance. In addition, the authors present a benchmark for real-world old photographs of human faces. Experiments demonstrate its advanced performance from perceptual quality and fidelity of restoration results.

**Questions:**

Please refer to the Strengths And Weaknesses!

**Ethical Concerns:**

["NO or VERY MINOR ethics concerns only"]

**Final Justification:**

Overall, I appreciate the author's efforts during the rebuttal and consider this a high-quality submission. Incorporating the aforementioned results will further strengthen the methodological rigor and reproducibility. Therefore, I am raising my score from Accept to Strong Accept.

**Limitations:**

Please refer to the Strengths And Weaknesses!

**Quality:**

4

**Strengths And Weaknesses:**

Paper strength: The authors propose a novel perspective by generating pseudo-labeled faces as references using a pre-trained diffusion model. This novel motivation is supported by comprehensive experiments, which validate the effectiveness of pseudo-labeled references. Additionally, paper writing on this paper is clear and easy to understand.
The design of selective guidance using pseudo-labels is interesting, making SSDiff support the region-specific stylized restoration, like hair and lips. Meanwhile, the proposed guidance strategy is effective, as demonstrated by extensive quantitative and qualitative experiments.

Major Weaknesses:
I have observed the pipeline of the method is mainly used for faded old photos with damage, and the three components of the method correspond to the blurred, damaged, and faded cases. However, old face photographs may not always have damage, and when this happens, is the method still applicable? Figure 6 demonstrates this to some extent, but can there be quantitative evidence?
The guidance process relies somewhat on face-parsing maps, but the robustness of the face-parsing maps is not explored.
When the face structure stabilizes, SSDiff guides coloring, but for old photos with different degrees of degradation, this moment T_1 should not be deterministic, it will be in a range, and the authors should clarify this.
In the Selective Restoration Guidance subsection, the authors do not describe how to select the breakage regions of images that do not contain the face component areas.
Minor Weaknesses:
The visualization shows good fidelity, but regarding the quantitative analysis of fidelity, why use CLIP-based face identity distances instead of normal identity distances?
The s = 3.5e-3 in Implementation Details does not match Figure 3; it should be S_s.

---

> ### Author Rebuttal · Authors · 2025-07-30
>
> Thank you for recognizing the novelty and rationality of our SSDiff framework, including the idea of pseudo-labeled references, the clear writing, and the effectiveness of the selective guidance strategy. Below, we address your concerns point by point. **As the current rebuttal system does not provide a way to include figures**, some clarifications are given quantitatively, and the corresponding visualizations will be added in the final version.
>
> * **Weak1: Quantitative evidence.** Yes, our method is applicable even when old photos have no visible breakage. When masks are empty, these modules act as normal restoration, and the later-stage color refinement still benefits faded images. **Table S1 in our supplementary** reports results on WebPhoto-Test (407 faces) and CelebA-Child (180 faces), both consisting of old face photos without breakage. SSDiff achieves the best or second-best scores across quantitative results, including FID and NIQE on the above benchmarks, confirming its effectiveness beyond broken cases with quantitative evidence.
>
> * **Weak2: Robustness of face-parsing maps.**
> The performance of SSDiff is generally robust to inaccuracies in the face parsing network. In our framework, the pre-trained parsing model is used only to provide a coarse directional signal during the reverse diffusion process, similar to the loose guidance mechanism in classifier-guided diffusion, rather than as a strict constraint. As long as this signal is not severely misleading (i.e., without gross parsing failures), the strong generative prior of the frozen diffusion model dominates the reconstruction and can recover plausible facial structures.
>
>     As discussed in **Sec. C.1 of the supplementary**, face parsing is considerably more robust than facial landmarks or 3D face models. Based on extensive experiments, we have observed that parsing maps almost never fail completely, even for degraded faces, and typically provide a sufficiently accurate spatial prior. To quantify this, we manually degraded 100 randomly selected high-quality CelebA‑HQ faces using the degradation model in GFPGAN [1], where degradation parameters (blur kernel σ, downsample scale r, Gaussian noise δ, JPEG quality factor q) are randomly sampled from (0.2:10, 12:16, 0:15, 60:100). These degraded images can be classified as hard-degraded following the standard of Fig. S1 in the supplementary, still achieve an average **face contours IoU of 79%** between their parsing-map and those of original high-quality images.
>
>     Qualitatively, across the visual results in the main paper, some parsing maps are clearly imperfect, yet the restored faces remain visually stable and of high quality. For example, **in Fig. 6 (second row)**, part of the face contour is occluded, leading to an inaccurate parsing map, but the final restoration remains natural, demonstrating that SSDiff does not rely on precise parsing boundaries.
>
>     To further evaluate robustness against parsing-map inaccuracies, we conducted a controlled experiment on the Medium subset of VintageFace by replacing the original parsing maps with pseudo-label maps generated with different guidance strengths $s$. The resulting face contours IoU with respect to the original parsing map are:
>
>     * for s=2.5e-4, IoU ≈ 76% (24% discrepancy);
>     * for s=1e-4, IoU ≈ 70% (30% discrepancy);
>
>     The resulting performance is summarized below:
>
>     |Parsing Maps|Ours|IOU(76%)|IOU(70%)|
>     |-|-|-|-|
>     |FID($\downarrow$)|128.3|131.1|133.4|
>     |MAN-IQA($\uparrow$)|0.395|0.391|0.382|
>     |Face Similarity ($\uparrow$)|1|0.985|0.955|
>
>     Here, Face Similarity (range [0, 1]) denotes the cosine similarity between features (extracted with ArcFace [3]) of the perturbed restoration and the original restoration (Ours). This result shows that errors in the parsing map network are not severe (Less than 40%), and the strong generative prior of the diffusion can propagate the correct cues to other face regions, preventing significant performance drops. This demonstrates that SSDiff is robust to these pre-trained networks. We will include these quantitative results and additional visual examples in the final version.
>
> * **Weak3: $T_1$ range.** Thank you for this suggestion. In extensive experiments on our whole benchmark, we also find that the transition point $T_1$ is not a fixed value but lies within a range. However, for simplicity and consistency across all experiments, we set $T_1=0.4T$, which corresponds well to most cases. In practice, depending on the degree of degradation, $T_1$ typically falls between $0.35T$ and $0.45T$; lighter degradations tend to shift $T_1$ slightly bigger. We will clarify this in the final version and include additional visual examples to illustrate the effect of this variation.
>
> * **Weak4: Selection of non-face breakage regions.** For selecting breakage regions outside facial components, our method combines the scratch mask with the labels from the face parsing map. The parsing map identifies non-face areas such as background, skin, and hair, while the scratch mask (binary: white=1 for breakage, black=0 for intact) localizes breakage regions. Taking the intersection of these two masks gives the breakage regions that do not contain facial components, which are then guided for restoration. We will incorporate these instructions into our final version.
>
> * **Weak5: Why CLIP-based face identity distances?** Traditional identity distances metrics rely on facial landmark detection (e.g., ArcFace [2]) and domain-specific face recognition embeddings. In input old photos with breakages, landmarks are often missing or heavily distorted, which leads to unstable or even invalid identity features and thus unreliable distances. To avoid this bias, we adopt CLIP-based face identity distances. CLIP embeddings are computed directly from the entire image without requiring landmark alignment, and they are trained on very diverse, cross-domain image-text data. This makes them more robust to blur, fading, and partial breakage, allowing us to measure identity similarity more stably and fairly under the challenging conditions of old-photo restoration. We will incorporate the above discussion into our final version.
>
> * **Weak6: Details error.** Thank you for your careful review. We will correct $s$ = 3.5e-3 to $s_s$ = 3.5e-3 in the implementation details of our main paper.
>
> [1] Wang X, Li Y, Zhang H, et al. Towards real-world blind face restoration with generative facial prior. CVPR 2021.
>
> [2] Deng J, Guo J, Xue N, et al. Arcface: Additive angular margin loss for deep face recognition. CVPR 2021.

---

> > ### Comment · Reviewer_pXcG · 2025-08-02
> > **Official Comment**
> >
> > After rebuttal, the authors have addressed my concerns. I have also carefully reviewed responses to other reviewers and found the questions well-addressed.
> > 1.	For Weak1, the experimental setup is comprehensive. The quantitative/qualitative results on two public benchmarks with old-photos in supplementary are persuasive.
> > 2.	Regarding additional dataset validation raised by other reviewers, the authors' responses are thorough. While DyPr and bLnM's proposed CelebA-HQ/LFW-Test validation with general degeneration could offer benefits to the paper, the validation is non-essential given the paper's focus on old-photo face restoration. The efficacy is well-demonstrated via private data and public benchmarks, with competitive performance on datasets with no old photos (CelebA-HQ/LFW-Test) indicating strong generalization without training, and preserving face identity well under synthetic degradations. That’s enough.
> > 3.	For Reviewer FoKK's suggestion, the verification results under diverse facial scenarios (e.g., poses, glasses) are convincing. Despite limited old-photos in these cases, I recommend including 1-2 visual examples in the final version for clarity.
> > 4.	For Weak2, the robustness analysis of parsing maps—including scratch detection, pseudo-labels, and style transfer—is well-supported by extensive results. These should be incorporated into the final version.
> > 5.	For Weak3-Weak6, supplementary explanatory notes should be added to the final version.
> > Overall, I appreciate the author's efforts during the rebuttal and consider this a high-quality submission. Incorporating the aforementioned results will further strengthen the methodological rigor and reproducibility.

---

> > > ### Author Response · Authors · 2025-08-05
> > >
> > > Thank you for the positive feedback. We are pleased that our response has addressed your concerns. We will carefully incorporate the reviewers’ comments and discussion points to further refine our revision.
> > >
> > > Once again, we sincerely appreciate your recognition of our paper.

---

> > ### Comment · Area_Chair_s5GD · 2025-08-07
> >
> > Dear Reviewer pXcG,
> >
> > This is a reminder that the author-reviewer discussion period is ending soon on Aug. 8 (AOE). Please submit the Mandatory Acknowledgement to confirm completion of this task.
> >
> > Thank you for your service in the review process.
> >
> > Area Chair

---

### Official Review · Reviewer_DyPr · 2025-06-27

**Clarity:** 3
**Significance:** 2
**Originality:** 2
**Rating:** 5
**Confidence:** 3

**Summary:**

This article presents a novel approach to old photo face restoration, termed Self-Supervised Selective-Guided Diffusion (SSDiff). The proposed method leverages a pre-trained diffusion model to generate pseudo-reference faces under weak guidance, which are then utilized to guide the restoration process through staged supervision. The method applies structural guidance throughout the denoising process and color refinement in later steps, aligned with the coarse-to-fine nature of diffusion. The experimental results demonstrate that SSDiff outperforms existing methods in terms of metrics such as FID, BRISQUE, TOPIQ and MAN-IQA. The qualitative comparison results also indicate that SSDiff is capable of producing natural colors, clear textures, and minimal artifacts, while maintaining high fidelity and identity consistency.

**Questions:**

1. Since the test set lacks ground truth (GT), I wonder how FID is evaluated. My understanding is that it's calculated with respect to the FFHQ dataset. In that case, is the FID computed on 300 images reliable?
2. The authors only conducted quantitative tests on their self-collected dataset, which lacks ground truth (GT), and non-reference metrics are often unreliable in many cases. It would be beneficial to test the quantitative results on some datasets with GT or to manually simulate degradation on FFHQ and evaluate the performance.
3. The authors utilize pre-trained face parsing and scratch detection networks. It is unclear whether the generated images will be substantially degraded if these networks are not accurate. Furthermore, the incorporation of these models may lead to a notable increase in inference time. Providing quantitative results on computational cost and inference time would be beneficial.
4. The efficacy of the proposed method is heavily dependent on the accuracy of the pseudo labels. Nevertheless, it is a known issue that diffusion models tend to produce inconsistent images under weak supervision. Does the model possess robustness to handle such situations?

**Ethical Concerns:**

["NO or VERY MINOR ethics concerns only"]

**Final Justification:**

I have carefully read the reviewers' comments and the authors' responses to them. The authors have clarified these concerns directly and provided rich and convincing experiments, so I have decided to increase my score.

**Limitations:**

yes

**Quality:**

3

**Strengths And Weaknesses:**

**Strengths**

This article's analysis of images generated under different guided intensities is insightful, and the idea of utilizing images generated under weak supervision as a reference for color and texture is interesting.

**Weaknesses**

The method has several limitations, including reliance on accurate pseudo-labels and pre-existing networks, and limited quantitative evaluation. See the question section for details.

---

> ### Author Rebuttal · Authors · 2025-07-30
>
> Thank you for highlighting the insightful analysis of guided intensities and the novel idea of using weakly supervised generated images as references for color and texture. We address your concerns point by point below. **As the current rebuttal system does not provide a way to include figures**, some clarifications are given quantitatively, and corresponding visualizations will be added in the final version.
>
> * **Q1: Calculation of FID.**
> The calculation of FID in our experiments follows official codes of VQFR [1] rather than computing statistics only on 300 images in our VintageFace. Specifically, we use the pre-trained inception weight released with VQFR, which is trained on 70k high-quality FFHQ images. FID is then computed between the feature distribution of our restored images and that of the reference FFHQ distribution. Therefore, the reliability of FID does not depend on the number of images in VintageFace itself, but on the fixed statistics from the large-scale FFHQ reference.
>
> * **Q2: Reference metrics on synthetic data.**
> Due to time constraints, we randomly select 1000 face images from CelebA-HQ for evaluation. Synthetic degradations are applied using the same degradation model as GFPGAN [2], where degradation parameters (blur kernel σ, downsample scale r, Gaussian noise δ, JPEG quality factor q) are randomly sampled from (0.2:10, 1:8, 0:15, 60:100) to form a **mixed-degradation** set. We report LPIPS and DISTS to assess perceptual quality and include face landmark distance (LMD) as a fidelity metric (as in VQFR [1]). PSNR and SSIM are not used as they primarily measure pixel similarity rather than face identity preservation.
>
>     |CelebA-HQ (synthetic)|DifFace|DiffBIR|DDNM|PGDiff|Ours|
>     |-|-|-|-|-|-|
>     |LPIPS ($\downarrow$)|0.2722|**0.2659**|0.4012|0.3104|*0.2695*|
>     |DISTS ($\downarrow$)|0.1441|*0.1431*|0.3281|0.1636|**0.1320**|
>     |LMD ($\downarrow$)|3.37|**3.01**|3.52 |4.15|*3.29*|
>
>     Compared with existing diffusion-based methods, our method achieves competitive performance, only behind DiffBIR, which is trained on large paired datasets generated with the same degradation model.  Note that our focus is on old-photo face restoration rather than blind face restoration. Nevertheless, as shown in the above tables and in our paper, SSDiff shows strong robustness to synthetic degradations. These results indicate that SSDiff is capable of handling both old face photos and synthetic degraded faces, whereas methods such as DiffBIR, though effective for synthetic degradations, struggle with old face photos.
>
>
> * **Q3_1: Robustness of pre-trained networks.**
> SSDiff is generally robust to inaccuracies in external components such as face parsing and scratch detection networks. These pre-trained networks provide only coarse directional guidance during reverse diffusion—similar to classifier-guided diffusion—rather than strict constraints. As long as their outputs are not severely erroneous, the strong generative prior of the frozen diffusion model dominates reconstruction and recovers plausible structures. Consequently, even when these auxiliary outputs are imperfect, the restorations remain stable and of high quality. Below, we analyze visual cases present in our paper to prove this point:
>
>     **Parsing maps:** As shown in **Fig. 6 (second row) of the main paper**, even with occluded contours, leading to inaccurate parsing, the results remain visually natural. Furthermore, as discussed in our analysis of "Weak2" for Reviewer pXcG, even under severe degradations, the IoU between degraded facial contours and GT contours remains around 79%, demonstrating the robustness of the parsing network.
>
>     **Scratch detection:** As shown in **Fig. S4 (second row) of our supplementary**, large stains on the head are only partially detected, yet the restoration remains natural. Similarly, **in the failure case in Fig. 11**, only a fraction of the stains are detected, but the restored results are still better than other methods. This demonstrates that the model uses detected regions as a rough guide and can roughly recover nearby features even with incomplete masks.
>
>     To quantify this robustness, we conduct additional experiments on the Medium degradation subset:
>
>     1. **Parsing maps:** We introduce inaccuracies by replacing the original parsing map with the pseudo-label parsing map of different strengths $s$, where the resulting errors are even larger than those observed in parsing maps under severe degradations (79% IOU). The resulting facial contours IoU with the original parsing maps is:
>         * for s=2.5e-4, IoU ≈ 76% (24% discrepancy);
>         * for s=1e-4, IoU ≈ 70% (30% discrepancy);
>
>         |Parsing Maps|Ours|IOU(76%)|IOU(70%)|
>         |-|-|-|-|
>         |FID($\downarrow$)|128.3|131.1|133.4|
>         |MAN-IQA($\uparrow$)|0.395|0.391|0.382|
>         |Face Similarity ($\uparrow$)|1|0.985|0.955|
>
>     2. **Scratch masks:** Randomly flip 10%, 20%, and 30% of the masks of breakage regions to simulate a situation where some of the breakages have not been detected.
>
>         |Scratch Masks|Ours|10% flip|20% flip|30% flip|
>         |-|-|-|-|-|
>         |FID($\downarrow$)|128.3|129.2|130.7|132.5|
>         |MAN-IQA($\uparrow$)|0.395|0.391|0.379|0.381|
>         |Face Similarity ($\uparrow$)|1|0.974|0.953|0.937|
>
>         Here, Face Similarity (range [0, 1]) denotes the cosine similarity between features (extracted with ArcFace [3]) of the perturbed restoration and the original restoration (Ours). This result shows that errors in the parsing map network are not severe (Less than 40%), and the strong generative prior of the diffusion can propagate the correct cues to other face regions, preventing significant performance drops. This demonstrates that SSDiff is robust to these pre-trained networks. We will include these quantitative results and additional visual examples in the final version.
>
>
> * **Q3_2: Computational Costs & Speed.**
> As detailed in **Sec. C.2 and Sec. C.3 of our supplementary**, we have analyzed the impact of the auxiliary components (e.g., pre-trained face parsing and scratch detection networks) on both computational cost and inference time:
>
>     **Params and FLOPs:** As shown in **Table S2 of our supplementary**, our SSDiff requires fewer parameters and FLOPs than existing diffusion-based methods. Because the face parsing and scratch detection networks can be executed offline, reported values in the supplementary do not include them. For completeness, if we include these components, the total becomes 105.4 M parameters and 140.6 G FLOPs, still significantly lower than DifFace (175.4 M, 268.8 G) and DiffBIR (1717 M, 23234 G). PGDiff also depends on scratch masks; using the same scratch detection model, its cost would be roughly 84.3 M parameters and 146.8 G FLOPs, comparable to ours. These results indicate that SSDiff does not suffer from a computational disadvantage.
>
>     **Inference Speed:** As reported in **Sec. C.2 of our supplementary**, all auxiliary networks are executed once in reverse diffusion rather than at every sampling step. Even when all networks run online, the added latency on an NVIDIA RTX 4090 is only about 95 ms, which is negligible compared to the tens of seconds required by the diffusion sampling itself.
>
>
> * **Q4: Robustness to pseudo-label inconsistency.**
> We appreciate your concern. This issue was carefully considered in the design of SSDiff. As shown in the statistical analysis in Fig. 2 of the main paper, by controlling the strength of weak supervision, we ensure that the pseudo-labels (generated by a frozen diffusion model) closely align with facial contours, with only small deviations (reflected in the reported IoU values). Moreover, these pseudo-labels are not used as pixel-level constraints; they serve only as coarse guidance for color style and smoothness in damaged regions, so moderate inaccuracies have limited influence on the final restoration.
>
>     To further assess robustness under less accurate pseudo-labels, we conducted an additional experiment on the Medium degradation subset using pseudo-labels generated with a smaller sampling step size (s=1e-4), where the average IoU of facial contours with the original input drops from 87% to 70%. As shown below, compared to the original pseudo-labels (s=1e-3), the resulting image quality and Face Similarity (defined in "Q3_1") degrade slightly. This confirms that SSDiff remains relatively robust even when pseudo-labels are partially less reliable.
>
>     |Pseudo-label quality|MAN-IQA($\uparrow$)|FID($\uparrow$)|Face Similarity($\uparrow$)|
>     |-|-|-|-|
>     |IOU:87% (Orginal)|0.395|128.3|1|
>     |IOU:70%|0.374|136.6|0.929|
>
> [1] Gu Y, Wang X, Xie L, et al. Vqfr: Blind face restoration with vector-quantized dictionary and parallel decoder. ECCV 2022.
>
> [2] Wang X, Li Y, Zhang H, et al. Towards real-world blind face restoration with generative facial prior. CVPR 2021.
>
> [3] Deng J, Guo J, Xue N, et al. Arcface: Additive angular margin loss for deep face recognition. CVPR 2021.

---

> > ### Comment · Reviewer_DyPr · 2025-08-01
> >
> > I appreciate the authors' efforts in the rebuttal.
> > + **Q1: Calculation of FID.** It seems my understanding is correct. However, the calculation of FID is done by computing the distribution distance between the set of inference results and the reference set. Its accuracy depends not only on the size of the reference set, but also on the size of the inference results set (otherwise the distribution wouldn't be accurate).
> > + **Q2: Reference metrics on synthetic data.** It appears that the method doesn't show significant improvement on synthetic datasets. It's worth noting that the paper only uses the private dataset proposed by the authors as the result for quantitative comparison.
> > + **Q3: Robustness of pre-trained networks & Computational Costs.** Although the visualization results are not available, the authors provide extensive quantitative experiments, which are convincing to me. I suggest adding relevant experiments and figures to the revised version.
> > + **Q4: Robustness to pseudo-label inconsistency.** The experimental data is convincing. I suggest adding relevant experiments and figures to the revised versionversion.
> >
> > In summary, based on my concerns regarding Q1 and Q2, I have decided to maintain the current score. Meanwhile, I look forward to the feedback from other reviewers and may take it into consideration when determining my final score.

---

> > > ### Author Response · Authors · 2025-08-01
> > >
> > > We sincerely thank the reviewer for additional feedback and acknowledging the validity of our analyses and results w.r.t Q3 and Q4. We will incorporate the additional quantitative and qualitative results as suggested. For Q1 and Q2, we would like to add further comments.
> > >
> > > * **Q1:** We agree the reliability of FID depends on both the size of the reference and the generated sets. Nevertheless, this metric is selected based on the common practice used in previous methods such as PGDiff [15], DiffBIR [10], and DifFace [9], where FID is computed on limited-size test sets (e.g. WebPhoto-Test with 407 images). Considering the FID reliability, we have provided another 4 metrics including BRISQUE, TOPIQ, MAN IQA, and NIQE to assess the performance of our method in a comprehensive manner, as shown in Table 1 in the main paper and Table S1 in the supplementary material. Therefore, we respectfully believe the evaluation results of our method are not biased.
> > >
> > > * **Q2:** Our evaluation **does not conduct on private data only**. As shown in Table S1 and Fig. S7 of the supplementary material, we report both quantitative and qualitative results on old photos (with blur, fade, without breakages) from the public datasets WebPhoto-Test [5] and CelebA-Child [5], where our method consistently achieves superior performance across datasets.
> > >
> > >     In addition, the blind‑degradation setting used to construct synthetic datasets is not representative of real old‑photo degradations, which are the primary focus of our method. DiffBIR performs slightly better in this synthetic setting because it is specifically trained on such paired synthetic data, while our method is designed for old‑photo degradation restoration with a training‑free pattern. Nevertheless, these results also demonstrate that our method remains competitive and robust in this general blind degradation. Therefore, we respectfully believe that the results on synthetic CelebA‑HQ shown in our rebuttal correspond to an unfair setting and do not diminish our contribution.

---

> > > > ### Comment · Reviewer_DyPr · 2025-08-02
> > > >
> > > > Thank you for the authors' response.
> > > >
> > > > Regarding Q1, this is also my main concern. In fact, non-reference quality metrics are generally not very reliable. I noticed that the reported FID scores are all above 100, which typically indicates that not all generated images are as visually appealing as the ones showcased in the paper. Other non-reference metrics also have limitations—for instance, NIQE scores can sometimes be lower when generated images contain more unrealistic details. This is one of the main reasons I encourage the authors to conduct evaluations on synthetic datasets. Of course, accurately simulating the degradation process of real old photographs remains an open and challenging problem, which is not the main focus of this work.
> > > >
> > > > Regarding Q2, I have read Table S1. First, comparisons on such important public datasets should be included in the main paper rather than the supplementary material. Second, I noticed that only FID and NIQE are reported. I recommend including additional quantitative metrics in the revised version to provide a more comprehensive evaluation, in line with the metrics used in the main paper.
> > > >
> > > > Overall, this is an interesting paper.

---

> > > > > ### Author Response · Authors · 2025-08-02
> > > > >
> > > > > We sincerely thank the reviewer for the detailed follow‑up and for acknowledging the effectiveness of SSDiff across additional benchmarks. We will include the results from Table S1, with extended metrics, in the main paper.
> > > > >
> > > > > * **For Q1:** We acknowledge the limitations of non-reference metrics. However, high FID values do not necessarily indicate poor visual quality but are primarily due to the limited size of evaluation datasets. As shown in Table-S1 in the supplementary, CelebA-Child (with far fewer images) yields a higher FID (>100) than WebPhoto-Test, even though WebPhoto-Test with more severe degradations (as described in the original paper [5] that introduced these two test sets) and consequently produces poorer restoration results. Therefore, we respectfully believe that dataset size, rather than restoration quality, is the primary reason for FID>100.
> > > > >
> > > > > Once again, we sincerely appreciate your thoughtful comments and the recognition of our work.

---

> > > > > > ### Comment · Reviewer_DyPr · 2025-08-06
> > > > > >
> > > > > > Thank you for the authors' response. My main concerns have been clarified (I also expect the authors to tackle the problem of non-reference quality metrics in the future and make more contributions to this field). I hope that in the camera-ready version, the authors will take into account the suggestions from reviewers FoKK, bLnM, DyPr, and pXcG and make revisions accordingly, and provide more visualization results (it's a pity that this year's rebuttal doesn't allow visual results to be shown). Additionally, I encourage the authors to open-source the relevant code. I've decided to increase my score to *Accept*.

---

> > > > > > > ### Author Response · Authors · 2025-08-09
> > > > > > >
> > > > > > > We sincerely thank your positive feedback and for increasing the score to Accept. We appreciate your constructive suggestions, especially regarding the incorporation of non-reference quality metrics in the future, the integration of feedback from other reviewers, and the encouragement to release our code. In the camera-ready version, we will carefully address these points and provide more visualization results to strengthen the presentation of our work.
> > > > > > >
> > > > > > > Once again, we sincerely appreciate your thoughtful comments and the recognition of our work.

---

### Official Review · Reviewer_bLnM · 2025-06-29

**Clarity:** 3
**Significance:** 2
**Originality:** 2
**Rating:** 4
**Confidence:** 4

**Summary:**

This paper introduces a novel framework, SSDiff, designed to address the unique challenges of restoring old face photographs, which typically suffer from complex and compounded degradations such as breakage, fading, and blur. The authors propose generating pseudo-references using a pre-trained diffusion model under weak guidance. These pseudo-labels, although not identity-preserving, exhibit realistic facial contours and natural color tones, which can effectively guide restoration. SSDiff applies selective restoration guidance early in the denoising process for structural and breakage recovery, and selective coloring guidance later for natural skin tone refinement. This aligns with the coarse-to-fine nature of diffusion models and avoids identity mismatches. The framework leverages face parsing maps and scratch masks to selectively apply supervision, avoiding over-correction in identity-sensitive regions (like eyes or mouth), while enhancing degraded areas like skin and background.

Also, the paper introduce VintageFace, a curated set of 300 real old face photos categorized into simple, medium, and hard degradation levels, enabling robust evaluation under diverse degradation conditions.

This paper also demonstrates strong qualitative and quantitative performance, outperforming existing GAN-based and zero-shot diffusion-based methods in perceptual quality (FID, BRISQUE, TOPIQ, MAN-IQA) and identity preservation (CLIP-based distance).

**Questions:**

- The proposed framework depends on multiple external models (face parsing, scratch detection, style transfer). How sensitive is SSDiff’s performance to the quality of these components?
- How is the 50 LQ selected for measuring fidelity of restoration?
- Have you considered evaluating SSDiff on existing open-source datasets related to degraded or historical face images (e.g., CelebA-HQ with synthetic degradations, or subsets from LFW with added artifacts)? Even a small-scale cross-dataset test or synthetic degradation benchmark could help demonstrate that the method generalizes beyond your self-built data.

**Ethical Concerns:**

["NO or VERY MINOR ethics concerns only"]

**Final Justification:**

The additional experiments and enriched dataset have addressed majority of my concerns. Thus I’m happy to adjust the score accordingly.

**Limitations:**

Yes.

**Quality:**

3

**Strengths And Weaknesses:**

Strengths:
- Innovative Self-Supervised Framework: The paper proposes SSDiff, a training-free framework that uses self-generated "pseudo-reference" faces to guide restoration, avoiding the need for paired training data.
- Staged Guidance: The model uses a staged guidance system, focusing first on structural restoration and then on color refinement in later steps, which aligns with the coarse-to-fine nature of diffusion models.
- Selective Restoration: By using face parsing maps and scratch masks, the method selectively restores damaged areas while avoiding alterations to identity-sensitive features like eyes and mouth.
- New Benchmark: The authors constructed "VintageFace," a new benchmark of 300 real old face photos with varied degradation levels, to evaluate performance.

Weaknesses:
- Limited Benchmark Scale: The newly introduced VintageFace benchmark, while a valuable contribution, is relatively small, with 300 images.
- Narrow Identity Preservation Test: The evaluation of identity preservation was conducted using CLIP-based distance on a select subset of 50 inputs with well-preserved features, which may not represent performance on more severely degraded images.
- Specialized Focus: The paper's scope is highly specialized for the compounded degradations common in old photographs, such as breakage, fading, and severe blurring. While the performance is superior, the real-world impact is limited since such degraded old photos are relatively rare.

---

> ### Author Rebuttal · Authors · 2025-07-30
>
> Thank you for recognizing the novelty of our SSDiff, including the self-supervised design using pseudo-references, the staged coarse-to-fine guidance strategy, and the selective restoration mechanism that preserves identity-sensitive regions. Below, we address your concerns point by point. **As the current rebuttal system does not provide a way to include figures**, some clarifications are given quantitatively, and the corresponding visualizations will be added in the final version.
>
> * **Weak1: Limited Benchmark Scale.**
> The VintageFace benchmark contains 300 images, but this benchmark is not a primary contribution; rather, it was introduced because no such dataset currently exists to provide an initial evaluation set for restoration performance under extreme old-photo degradations, including breakage, fading, and severe blur. We plan to further expand this benchmark in future work. Moreover, as shown in **Table S1 in our supplementary**, we have also validated SSDiff on broader datasets containing old photos (WebPhoto-Test and CelebA-Child), demonstrating that our method is not limited by the current scale of VintageFace.
>
> * **Weak2: Narrow Identity Preservation Test.**
> To address your concern, we evaluate on the subset of CelebA-HQ (1000 images) with synthetic degradations generated following the degradation model in GFPGAN [1], where degradation parameters (blur kernel σ, downsample scale r, Gaussian noise δ, JPEG quality factor q) are randomly sampled from (7:9, 8:16, 20:40, 30:40) to form a **severe-degradation set** (measured using Fig. S1 in the supplementary). We recompute CLIP-based identity distances on this dataset, and we further adopt face landmark distance (LMD) [2] as the face fidelity metric. PSNR or SSIM are not used as they primarily measure pixel consistency rather than face identity preservation. As shown in the table below, our method remains consistent with the paper's findings: SSDiff keeps competitive among diffusion-based methods, only behind DiffBIR on the LMD metric. Note that DiffBIR is trained on large paired datasets synthesized using the same degradation model, whereas SSDiff is training-free.
>
>     |Degradation Type|DifFace|DiffBIR|DDNM|PGDiff|Ours|
>     |-|-|-|-|-|-|
>     |Identity Distance ($\downarrow$)|0.171|*0.154*|0.187|0.202|**0.153**|
>     |LMD ($\downarrow$)|4.20|**3.88**|4.85|5.19|*4.12*|
>
>
> * **Q1: Robustness of pre-trained networks.**
> SSDiff is generally robust to inaccuracies in external components (face parsing, scratch detection, style transfer). These networks only provide coarse directional signals during reverse diffusion, similar to classifier-guided diffusion, and are not strict constraints. As long as the guidance is not severely misleading, the strong generative prior of the frozen diffusion model dominates reconstruction. Across many visual results in the paper, there are inevitably cases where these networks are imperfect, yet the restorations remain stable and of high quality.
>
>     **Parsing maps:** As shown in **Fig. 6 (second row) in our main paper**, even with occluded contours leading to inaccurate parsing, the results remain visually faithful. Furthermore, as discussed in our analysis of "Weak2" for Reviewer pXcG, even under severe degradations, the IoU between degraded facial contours and GT contours remains around 79%, demonstrating its robustness.  In addition, the results of the parsing map do not directly supervise the guidance, but rather in an indirect manner, also leading to better robustness.
>
>     **Scratch detection:** As shown in **Fig. S4 (second row) in our supplementary**, large stains on the head are only partially detected, yet the restoration remains natural. Similarly, **in the failure case in Fig. 11**, only a fraction of the stains are detected, but the restored results are still better than other methods. This demonstrates that the model uses detected regions as a rough guide and can roughly recover nearby structures even with incomplete masks.
>
>     **Style transfer:** Although the main paper does not include intermediate style-transfer outputs, we observe the intermediate style‑transfer result for Fig. 7 (first row). The result shows that transferred images initially have a yellowish facial tint; however, once used as a coarse color prior, the diffusion process refines and propagates realistic colors, producing a natural complexion.
>
>     To quantify this robustness, we conducted additional experiments on the Medium type subset:
>
>     1. **Parsing maps:** We introduced inaccuracies by replacing the original parsing maps with pseudo-label parsing maps of different strengths $s$, where the resulting errors are even larger than those observed in parsing maps under severe degradations (79% IOU). The resulting IoU with the original parsing map is:
>         * for s=2.5e-4, IoU ≈ 76% (24% discrepancy);
>         * for s=1e-4, IoU ≈ 70% (30% discrepancy);
>
>         |Parsing Maps|Ours|IOU(76%)|IOU(70%)|
>         |-|-|-|-|
>         |FID($\downarrow$)|128.3|131.1|133.4|
>         |MAN-IQA($\uparrow$)|0.395|0.391|0.382|
>         |Face Similarity ($\uparrow$)|1|0.985|0.955|
>
>     2. **Scratch masks:** Randomly flip 10%, 20%, and 30% of the masks of breakage regions to simulate a situation where some of the breakages have not been detected.
>
>         |Scratch Masks|Ours|10% flip|20% flip|30% flip|
>         |-|-|-|-|-|
>         |FID($\downarrow$)|128.3|129.2|130.7|132.5|
>         |MAN-IQA($\uparrow$)|0.395|0.391|0.379|0.381|
>         |Face Similarity ($\uparrow$)|1|0.974|0.953|0.937|
>
>     3. **Style transfer:** We weaken the style transfer guidance by reducing the style factor α from 0 to 0.1 and 0.2, slightly affecting color and content.
>
>         |Style Transfer|Ours (α=0)|Poor-1 (α=0.1)|Poor-2 (α=0.2)|
>         |-|-|-|-|
>         |FID($\downarrow$)|128.3|129.2|128.8|
>         |MAN-IQA($\uparrow$)|0.395|0.396|0.392|
>         |Face Similarity ($\uparrow$)|1|0.977|0.959|
>
>     Here, Face Similarity (range [0, 1]) denotes the cosine similarity between features (extracted with ArcFace [3]) of the perturbed restoration and the original restoration (Ours). **These results show that errors in the pre-trained networks are not severe (Less than 40%), and the strong generative prior of the diffusion model can propagate the correct cues to other regions, preventing significant performance drops.** This demonstrates that SSDiff is robust to these pre-trained networks. We will include these quantitative results and additional visual examples in the final version.
>
> * **Q2: How to select 50 LQ samples for measuring fidelity.**
> The 50 LQ images used for fidelity evaluation are chosen from samples with relatively mild degradation, particularly those without breakage or severe blur in key facial components such as eyes, nose, and mouth. Since in the absence of ground truth, identity distances have to be computed with input old photos. For heavily degraded images with severe blur, fading, or breakage, a successful restoration even incorrectly increases the identity distance (where a smaller distance indicates better fidelity) because the restoration corrects degradations that are inadvertently treated as part of the original face identity. Therefore, to mitigate this bias, we select old photos with lighter degradation and better-preserved facial features, providing a more reliable evaluation of identity fidelity.
>
>
> * **Weak3 & Q3: Generalization to other datasets and degradations.**
> As shown in **Table S1 and Fig. S7 of our supplementary**, we have evaluated SSDiff on popular benchmarks beyond our VintageFace, including WebPhoto-Test (407 LQ faces) and CelebA-Child (180 LQ faces). These sets contain a large number of historical face images with varying degrees of blur and fading but no breakage, where our masks automatically degenerate to standard restoration (e.g., all black). SSDiff achieves superior quantitative (FID, NIQE) and qualitative results on both benchmarks, demonstrating strong generalization.
>
>     Following your valuable suggestion, we further performed additional evaluations on a 1000-image subset of CelebA-HQ using the same degradation model in GFPGAN [1], where degradation parameters (blur kernel σ, downsample scale r, Gaussian noise δ, JPEG quality factor q) are randomly sampled from (0.2:10, 1:8, 0:15, 60:100) to form a **mixed-degradation set**. We further evaluate on the real benchmark you mentioned (LFW-Test with 1711 LQ faces).
>
>     |CelebA-HQ (synthetic)|DifFace|DiffBIR|DDNM|PGDiff|Ours|
>     |-|-|-|-|-|-|
>     |LPIPS ($\downarrow$)|0.2722|**0.2659**|0.4112|0.3104|*0.2695*|
>     |DISTS ($\downarrow$)|0.1441|*0.1431*|0.3281|0.1636|**0.1320**|
>     |FID ($\downarrow$)|69.3|*51.2*|115.8|58.1|**50.3**|
>     |NIQE ($\downarrow$)|*4.29*|6.08|8.83|4.837|**4.16**|
>
>     |LFW-Test (real)|DifFace|DiffBIR|DDNM|PGDiff|Ours|
>     |-|-|-|-|-|-|
>     |FID ($\downarrow$)|46.4|**40.4**|112.3|47.3|*42.2*|
>     |NIQE ($\downarrow$)|4.30|5.74|9.64|*4.11*|**4.03**|
>     |MAN-IQA ($\uparrow$)|0.368|**0.423**|0.275|0.373|*0.411*|
>
>     As shown in these tables, SSDiff consistently achieves the best or second-best results across both reference and non-reference metrics. Note that our focus is on old-photo face restoration rather than real-world face restoration. Nevertheless, as shown in the above tables and in our paper, SSDiff demonstrates strong robustness to real-world degradations. These results indicate that SSDiff is capable of handling both old face photos and real-world degraded faces, whereas methods such as DiffBIR, though effective for real degradations, struggle with old face photos.
>
> [1] Wang X, Li Y, Zhang H, et al. Towards real-world blind face restoration with generative facial prior. CVPR 2021.
>
> [2] Gu Y, Wang X, Xie L, et al. Vqfr: Blind face restoration with vector-quantized dictionary and parallel decoder. ECCV 2022.
>
> [3] Deng J, Guo J, Xue N, et al. Arcface: Additive angular margin loss for deep face recognition. CVPR 2021.

---

> > ### Comment · Reviewer_bLnM · 2025-08-06
> >
> > Thanks authors for the detailed rebuttal, the additional experiments and enriched dataset have addressed majority of my concerns. Thus I’m happy to adjust the score accordingly.

---

> > > ### Author Response · Authors · 2025-08-06
> > >
> > > Thank you for your positive feedback. We are glad our response has addressed your concerns, and we will carefully incorporate your comments and discussion points to further improve the revision. Once again, we sincerely appreciate your recognition of our work.

---

### Official Review · Reviewer_FoKK · 2025-07-03

**Clarity:** 3
**Significance:** 3
**Originality:** 3
**Rating:** 4
**Confidence:** 5

**Summary:**

This paper introduces SSDiff, a training-free framework designed to address specific degradations in old face photos, such as breakage and color fading. SSDiff leverages a pre-trained face diffusion model to generate a pseudo-label, which encapsulates both structural and color information. Based on this pseudo-label, a staged, region-specific guidance scheme is proposed to enhance the restoration process. Additionally, a real-world benchmark, VintageFace, is introduced, and SSDiff demonstrates superior performance compared to existing methods on this dataset.

**Questions:**

1. The caption for Figure 3 should be revised for clarity and precision.
2. It is unclear what \hat{x_{t}} represents.

**Ethical Concerns:**

["NO or VERY MINOR ethics concerns only"]

**Final Justification:**

After reading the rebuttal and  other reviewers' opinion, most of my concerns have been addressed.

The authors are suggested to include the rebuttal's results and explanations.

**Limitations:**

yes

**Quality:**

3

**Strengths And Weaknesses:**

Strengths：
1. The idea of the paper is straightforward: it uses a clean pseudo-label to guide the inverse process. Additionally, the staged approach proposed by the authors for handling different types of information is quite reasonable. These concepts are supported by analytical results presented in Fig. 2 and Fig. 4, making the overall logic sound.
2. From both quantitative and qualitative results, SSDiff demonstrates good performance on the VintageFace benchmark.

Weaknesses：
1. The experimental settings and scenarios are somewhat incomplete:
	a. The results in Table 2 are quite puzzling. Without incorporating the self-supervised strategy proposed by the authors, this experimental setup achieves near state-of-the-art performance. Please provide corresponding explanations and clarifications.
	b. The paper only presents results for front-facing photos, without showing results for other poses, such as profiles. Additionally, the paper does not include results for the hard-case photos introduced in the VintageFace benchmark. Please clarify how SSDiff performs on these two common cases, as well as on cases involving individuals wearing glasses.
	c. Are the results in the table of Fig. 8 based on a single image? If so, the selection of these two scale parameters may not be representative.
	d. In many of the loss calculations, the authors exclude the eyes, nose, and mouth. How does SSDiff perform if these areas contain significant scratches or degradation?
	e. How does the author ensure identity preservation? There doesn’t appear to be any constraint on identity during the supervision process.
	f. The handling of the background regions is suboptimal (see Fig. 5, Fig. 9, and Fig. S4, third case).

2. Some of the claims are puzzling and may not be entirely accurate.
	a. What is the author attempting to convey in lines 191-195? The equation presented appears to be an optimization formulation, but in the experiments, s_{w}sw​ and s_{s}ss​ are empirically chosen constants.
	b. In line 241, the authors claim that the face parsing network is robust to degradations. I suggest that the authors include results for some hard cases, such as those with significant degradations or profile poses, to better support this claim.
	c. In line 38 of supp, Why does the style transfer model only need to be applied in a single step?	a. The caption for Figure 3 should be revised for clarity and precision.
 It is unclear what \hat{x_{t}} represents.

---

> ### Author Rebuttal · Authors · 2025-07-30
>
> Thank you for recognizing the clear idea of our method, the staged guidance design, and the soundness of our analysis, as well as the strong performance of SSDiff on benchmarks. We also appreciate your suggestions for adding more visual examples. **While the rebuttal system does not allow us to include additional figures**, we provide quantitative clarifications where possible and will incorporate your recommended visualizations in our final version.
>
> * **Weak1 (a): Explanation of Table 2.**
> The baseline without our self-supervised strategy still benefits from two factors: 1. We retain the L1 loss, which provides a coarse directional signal that makes restorations visually clearer. 2. Mask-guided diffusion explicitly marks unreliable breakages so that frozen diffusions can use its generation to hallucinate missing content. Although this can only recover simple breakages, it naturally leads to better results than methods without any breakage repair. However, these results should not be confused with the self-supervised strategy. By introducing pseudo-reference guidance, it substantially improves color, texture, and fine details.
>
> * **Weak1 (b): Results in more scenarios.**
> Thank you for this suggestion, which makes our evaluation more comprehensive. Below, we evaluate our SSDiff with existing diffusion-based methods:
>
>     **More poses:** We collect 20 relatively profile-view faces (fully side profiles are rare) from VintageFace and the web, and 20 frontal faces with comparable degradation levels (measured using Fig. S1 in the supplementary). The metrics show no significant gap between the two settings, demonstrating that such cases do not affect the robustness.
>
>     |Poses|Front|Profile-view|
>     |-|-|-|
>     |FID($\downarrow$)|128.6|132.9|
>     |MAN-IQA($\uparrow$)|0.3975|0.3920|
>
>     Furthermore, we evaluate 100 fully profile-view faces from real benchmarks (LFW-Test, WebPhoto-Test, Wider-Test); SSDiff surpasses DiffBIR, demonstrating generalization to pose variations. (DiffBIR has shown strong profile-view visual results in its paper.)
>
>     |Real profile-view|DifFace|DiffBIR|PGDiff|Ours|
>     |-|-|-|-|-|
>     |FID($\downarrow$)|151.5|*146.0*|158.1|**143.5**|
>     |NIQE($\downarrow$)|*4.116*|5.803|6.225|**4.035**|
>     |MAN-IQA($\uparrow$)|0.3603|*0.3912*|0.3712|**0.3955**|
>
>     **Hard cases:** Some visualizations already come from hard-degradation subsets (e.g., **first row in Fig. 1, the third row in Fig. 5, and the second row in Fig. 6 in our main paper**). We understand that this question may refer to more extreme degradations. Therefore, we evaluate on the first 200 faces of the real benchmark Wider-Test (severely degraded but no breakages); SSDiff remains competitive.
>
>     |Severe degradation|DifFace|DiffBIR|PGDiff|Ours|
>     |-|-|-|-|-|
>     |FID($\downarrow$)|*82.60*|82.75|91.80|**81.02**|
>     |NIQE($\downarrow$)|*4.403*|5.891|4.930|**4.092**|
>     |MAN-IQA($\uparrow$)|0.3386|**0.3781**|0.3415|*0.3697*|
>
>     **Wearing glasses:** Due to scarcity, we collect 10 pairs of old-photo faces (glasses vs. non-glasses) with similar degradation levels from the web; the performance gap is negligible, demonstrating that such cases do not affect the robustness.
>
>     |Glasses|Wear|Not wear|
>     |-|-|-|
>     |FID($\downarrow$)|128.4|127.6|
>     |MAN-IQA($\uparrow$)|0.3838|0.3814|
>
>     We further evaluate 100 faces with glasses from real benchmarks (LFW-Test, WebPhoto-Test, and Wider-Test), where SSDiff performs competitively with only a minor gap to DiffBIR, which is built upon a fine‑tuned SD 2.1 model with heavy training costs, while our method is training-free and inference on a frozen DDPM generator.
>
>     |Real data (wear glasses)|DifFace|DiffBIR|PGDiff|Ours|
>     |-|-|-|-|-|
>     |FID($\downarrow$)|136.3|**119.8**|126.9|*121.7*|
>     |NIQE($\downarrow$)|*3.871*|5.567|5.241|**3.843**|
>     |MAN-IQA($\uparrow$)|0.3558|**0.4020**|0.3284|*0.3962*|
>
>     Finally, VintageFace mainly consists of front-facing photos because decades ago, portraits were usually taken in studios to clearly record one's appearance, so profile views are rare. Similarly, glasses were uncommon at that time, so faces with glasses are rare. Therefore, our results don't contain examples of this. We will add these results in the final version.
>
> * **Weak1 (c): Clarification on Fig. 8.**
> All ablations, including those summarized in Fig. 8, are conducted on our Medium subset, which we consider representative because it spans a wide range of degradations from mild to near-severe. The single input image shown in the figure is only for illustration due to layout constraints and may have caused confusion. We will make this clear in the final version.
>
> * **Weak1 (d): Performance when eyes, nose, and mouth are with heavy scratches.**
> As shown in **Fig. 3 and in the second and third rows of Fig. 5 in the main paper**, SSDiff is able to recover scratches around eyes, nose, and mouth with natural visual quality. To validate more heavy scratches, we select 50 moderate‑scratch faces and 50 severe‑scratch faces of these regions from VintageFace. We evaluate the mean smoothness (e.g., Edge strength) before and after restoration. The table shows that although severe scratches produce poorer smoothness in inputs of these regions, smoothness of restorations with no significantly degraded areas, demonstrating the robustness of our SSDiff for eyes, nose, and mouth. This robustness comes from generative priors of diffusion, which use contextual facial cues and pseudo-reference guidance to recover these regions. Visualization can also prove this, in Fig. 11 of the main paper, where even heavily degraded, eyes, nose, and mouth are recovered relatively well.
>
>     |Type|Medium scratches|Severe scratches|
>     |-|-|-|
>     |Edge Strength($\downarrow$)|Input:114/Restored:46|Input:140/Restored:51|
>
> * **Weak1 (e): Identity preservation.**
> Identity consistency is preserved by combining the fidelity signal from a restorer with diffusion guidance. The restorer first produces a coarse but aligned structure and texture, and in reliable regions without breakages, this output is close to the original identity. These reliable regions are enforced through our L1 loss in all reverse diffusion processes, so that the guidance passed to diffusion models retains the original appearance, preventing identity drift in generation.
>
>     we avoid explicit constraints like landmarks or 3D face shape because such constraints are highly sensitive to errors and even fail under moderate degradations. Instead, our design follows the same philosophy as generation-based restoration methods for controlling fidelity: using a reliable restoration module to preserve original facial textures as a constraint. However, the difference is that we then use diffusion guidance rather than training optimisation to refine images. This avoids the need for end-to-end training while effectively constraining the diffusion so that the results do not drift away from the true identity.
>
> * **Weak1 (f): Handling of background regions.**
> The main reason for the less satisfactory background restoration is that the detected scratch fails to fully cover all breakage areas, and the diffusion itself has relatively weak generative ability for backgrounds compared to faces. Nevertheless, resulting backgrounds remain visually reasonable, showing the robustness of SSDiff for such detection error. We appreciate your observation, which indeed points out a potential direction for future work; we plan to explore region-adaptive guidance, where different areas (e.g., face vs. background) are guided with adaptive strengths to address this issue.
>
> * **Weak2 (a): Line 191-195.**
> The current formulation may cause a misunderstanding. The intention of Eq. 9 was not to suggest that $s_w$ and $s_s$ are learned during network optimization. We wish to express that empirically determining these two coefficients by manual search can minimize the overall guidance loss, achieving optimal restoration. In the final version, we will revise Eq. 9 to make clear that these are fixed hyperparameters rather than parameters obtained through optimization.
>
> * **Weak2 (b): Robustness of Face Parsing.**
> Our claim is based on extensive observations: under heavy degradations, facial landmarks and 3D face estimation often fail completely, whereas parsing networks still produce approximate facial contours. This is also supported by our analysis in "Weak2" of Reviewer pXcG, where the IoU value between severely degraded and GT facial contours is high (79%), and we further show that SSDiff remains robust even with imperfect parsing maps in that response. We will add more visual examples to better support this.
>
> * **Weak2 (c): Single-step use of style transfer.**
> Style transfer is applied only once, at the moment when face structures become relatively stable ($T=T_1$). The color distribution obtained at this moment provides a coarse but reliable color reference, which is then propagated and refined by the diffusion during the remaining reverse steps. Because the Selective Restoration Guidance is applied throughout, this single-step color initialization doesn't harm structural fidelity. Repeating style transfer at every diffusion step offers little additional benefit while incurring prohibitive computational cost ($T_1$ times longer runtime), so a single application is enough.
>
> * **Weak2 (d) & Q1 & Q2: Unclear of $\hat{x}_{t}$.**
> In Fig. 3, **$\hat{x}_{t}$** denotes the intermediate restored estimate of clean images at timestep t in the reverse diffusion. Specifically, at each step, the model predicts a clean image from the current noisy state (as formulated in Line 7 of Algorithm 1 in our main paper), and **$\hat{x}_{t}$** is this predicted clean sample. It indicates how images progressively become accurate as reverse diffusion proceeds. We will revise the figure caption in the final version.

---

> > ### Comment · Area_Chair_s5GD · 2025-08-07
> >
> > Dear Reviewer FoKK,
> >
> > This is a reminder that the author-reviewer discussion period is ending soon on Aug. 8 (AOE), and you have not yet responded to the authors' rebuttal. Please read the authors' rebuttal as soon as possible, engage in any necessary discussions, and consider if you would like to update your review and score. Please submit the Mandatory Acknowledgement to confirm completion of this task.
> >
> > Thank you for your service in the review process.
> >
> > Area Chair

---

> > ### Comment · Reviewer_FoKK · 2025-08-07
> >
> > Thanks for the authors' response.
> > Most of my concerns has been addressed.  The authors have also provided more supplementary experiments and explanations.
> > I tend to increase the score.
> > The authors are suggested to include the rebuttal's results and explanations.

---

> > > ### Author Response · Authors · 2025-08-09
> > >
> > > Thank you for your positive feedback. We are glad our response has addressed your concerns, and we will carefully incorporate your comments and discussion points to further improve the revision. Once again, we sincerely appreciate your recognition of our work.

---

### Comment · Area_Chair_s5GD · 2025-08-04

Dear Reviewers,

The authors have provided substantial additional quantitative results to address the reviewer's concerns.

Please review the authors' detailed responses. For those who haven't responded, please share whether the rebuttals address your concerns.

Please engage actively during this discussion period to help reach consensus or identify remaining disagreements.

Best regards, Area Chair

---

### Decision · Program_Chairs · 2025-09-17

**Decision:**

Accept (poster)

**Comment:**

This paper presents a training-free framework for old-photo face restoration using pseudo-reference faces from weak diffusion guidance with staged supervision (structural then color) and selective restoration. Reviewers praised the novel approach, technical soundness, strong results on the VintageFace benchmark, and clear presentation. Reviewers' initial concerns about robustness, limited evaluation scenarios, and computational costs were addressed during the rebuttal through experiments showing the algorithm's robustness to 30-40% parsing errors and better performance across diverse conditions. The rebuttal led to score increases, with all reviewers satisfied. Given the technical soundness, comprehensive evaluation, and consensus positive reviews, the AC recommends Acceptance.